# A Simple Temperature-Based Method to Estimate Heterogeneous Frozen Ground within a Distributed Watershed Model

Michael L. Follum[1,2], Jeffrey D. Niemann[2], Julie Parno[3], and Charles W. Downer[1]

[1]Coastal and Hydraulics Laboratory, Vicksburg, MS, 39180, USA.
[2]Department of Civil Engineering, Colorado State University, Fort Collins, CO, 80523, USA.
[3]Cold Regions Research and Engineering Laboratory, Hanover, NH, 03755, USA.

*Correspondence to*: Michael L. Follum (Michael.L.Follum@usace.army.mil)

## Abstract

Frozen ground can be important to flood production and is often heterogeneous within a watershed due to spatial variations in the available energy, insulation by snowpack and ground cover, and the thermal and moisture properties of the soil. The widely-used Continuous Frozen Ground Index (CFGI) model is a degree-day approach and identifies frozen ground using a simple frost index, which varies mainly with elevation through a temperature-elevation relationship. Similarly, snow depth and its insulating effect are also estimated based on elevation. The objective of this paper is to develop a model for frozen ground that (1) captures the spatial variations of frozen ground within a watershed, (2) allows the frozen ground model to be incorporated into a variety of watershed models, and (3) allows application in data sparse environments. To do this, we modify the existing CFGI method within the Gridded Surface Subsurface Hydrologic Analysis watershed model. Among the modifications, the snowpack and frost indices are simulated by replacing air temperature (a surrogate for the available energy) with a radiation-derived temperature that aims to better represent spatial variations in available energy. Ground cover is also included as an additional insulator of the soil. Furthermore, the modified Berggren Equation, which accounts for soil thermal conductivity and soil moisture, is used to convert the frost index into frost depth. The modified CFGI model is tested by application at six test sites within the Sleepers River Experimental Watershed in Vermont. Compared to the CFGI model, the modified CFGI model more accurately captures the variations in frozen ground between the sites, inter-annual variations in frozen ground depths at a given site, and the occurrence of frozen ground.

## 1 Introduction

Frozen ground (also known as frozen soil or soil frost) is important to predicting stormflows produced by certain watersheds (Shanley and Chalmers, 1999; McNamara et al., 1997; Prèvost et al., 1990; Woo, 1986). Several plot-scale studies have shown that frozen ground can impede infiltration and thus enhance runoff (Bayard et al., 2005; Dunne and Black, 1971; Stähli et al., 1999). Several of these studies have also shown that frozen ground is highly-variable temporally and spatially (Campbell et al., 2010; Shanley and Chalmers, 1999; Stähli, 2017), which affects the amount and type of runoff

(Wilcox et al., 1997). The presence, spatial pattern, and depth of frozen ground are driven by mass (water) and energy balances. The energy available from the atmosphere to thaw the soil is subject to the insulation of the snowpack (Pearson, 1920; Willis et al., 1961) and ground cover including any vegetation, woody debris, and leaf litter (Brown, 1966; Diebold, 1938; Fahey and Lang, 1975; Sartz, 1973; Stähli, 2017). MacKinney (1929) found that ground cover reduced the depth of frost penetration by 40% at a test site in Connecticut. Additionally, the presence and depth of frozen ground is affected by soil moisture (Fox, 1992; Willis et al., 1961) and the thermal conductivity of the soil (Farouki, 1981; Johansen, 1977).

Frozen ground has proven difficult to simulate within hydrologic models due to complex interactions of energy and water between the atmosphere, snowpack, and soil (Dun et al., 2010; Kennedy and Sharratt, 1998; Lin and McCool, 2006). Physically-based models of frozen ground, such as the Simultaneous Heat and Water (SHAW) model (Flerchinger and Saxton, 1989), the coupled heat and mass transfer model for soil-plant-atmosphere systems (COUP) (Jansson 2001; Jansson and Karlburg, 2010), and the Distributed Water-Heat Coupled (DWHC) model (Chen et al., 2007) have large parameter and forcing data requirements – such as wind speed, relative humidity, and short- and long-wave radiation – which restricts their applicability in many watershed. Additionally, these types of models either include, or are tightly coupled to soil moisture models, which can limit their applicability in models that do not explicitly simulate soil moisture content. To reduce data and parameter requirements and increase applicability, simple temperature-index or degree-day methods (Molnau and Bissell, 1983; Rekolainen and Posch, 1993) remain widely used within watershed models, including LISFLOOD (De Roo et al., 2001; Van Der Knijff et al., 2010), CREAMS (Rekolainen and Posch, 1993), and the Gridded Surface Subsurface Hydrologic Analysis (GSSHA) model (Downer and Ogden, 2004). Degree-day approaches typically accumulate the daily average temperature as a frost index (°C-days). When the frost index exceeds a threshold, the soil is considered frozen and impermeable to infiltration. The sudden restriction on infiltration can be an incorrect assumption, especially in forested environments where frozen soils often still experience infiltration (Lindstrom et al. 2002; Nyberg et al., 2001; Shanley and Chalmers, 1999). A limitation of degree-day approaches is that they are often untested against observed frost data because the frost index is not a physical property that can be compared to measurements. However, degree-day methods have been successful in capturing increased runoff from frozen ground events (Molnau and Bissell, 1983), and higher frost index values have been shown to correlate to deeper frost depths (Vermette and Christopher, 2008; Vermette and Kanack, 2012). Spatial variations of frozen ground within degree-day methods are typically based on variations in temperature (which are estimated from an elevation-temperature relationship) and variations in snowpack insulation (which are also typically inferred from an elevation-temperature relationship). Such reliance on elevation may lead to errors because Stähli (2017) found no clear connection between elevation and presence of frozen ground at test sites in the Swiss pre-alpine zone.

The objective of this paper is to develop a model for frozen ground that (1) captures the spatial variations of frozen ground within a watershed, (2) allows the frozen ground model to be incorporated into a variety of watershed models, and (3) allows application in data sparse environments where limited forcing data may prohibit use of energy balance methods. In this paper, we use the GSSHA watershed model and develop the frozen ground model by modifying the commonly used Conceptual Frozen Ground Index (CFGI) (Molnau and Bissell, 1983) method in four ways. First, the CFGI method is coupled to an

improved snowpack model that more accurately captures the spatial heterogeneity of the snowpack. In past applications of GSSHA, the CFGI method has been coupled with a temperature-index (TI) snowpack model based on SNOW-17 (Anderson, 1973; Anderson, 2006). However, Follum et al. (2015) proposed a Radiation-derived Temperature Index (RTI) snow model that uses a proxy temperature instead of air temperature to represent the energy available to the snowpack. Compared to the
TI model, the RTI model more directly includes the effects of shortwave radiation and canopy cover and was shown to better represent the spatial variations of snow cover and snow water equivalent (SWE) in the Senator Beck Basin in Colorado. The RTI model is adopted to simulate the snowpack in the present study. Second, the effects of shortwave radiation and canopy cover are included in the CFGI model when calculating the energy available at the snow or ground surface. These effects are included by using a similar radiation-derived proxy temperature when calculating the frost index. Third, the insulation effects
of ground cover are included by modifying the frost index equation. Fourth, an option is included to compute frost depth as a function of the frost index value. The modified Berggren Equation and similar Stefan Equation have been previously used to estimate frost depth from degree-days (Carey and Woo, 2005; DeWalle and Rango, 2008; Fox, 1992; Woo et al., 2004); a similar approach is used here to convert the frost index to frost depth.

      The following sections first describe the existing TI and CFGI models within GSSHA. The combination of these two
models serves as the baseline or control case for the experiments. Then, the RTI snow model and the modified CFGI frozen ground model (referred to as modCFGI) are described. Finally, the results of the TI/CFGI model and RTI/modCFGI models are compared to each other and to observations of snow depth, SWE, and frost depth at the Sleepers River Experimental Watershed (SREW) in Vermont.

## 2 Methodology

**2.1 TI Snowpack Model**

      The TI snow model was implemented into GSSHA by Follum et al. (2014), who provides additional information about the model. Although GSSHA allows a variable time step for multiple processes, it always uses an hourly time step ($\Delta t$) for snow calculations. GSSHA utilizes a structured grid in which each cell can have a different air temperature $T_a$ (°C) and precipitation $P$ (m h$^{-1}$). Air temperature is the primary driver of snowpack dynamics in the TI model and is estimated as:

$T_a = T_g + \emptyset\big(E_g - E_c\big),$                                             (1)

where $T_g$ (°C) is the air temperature at a gage, $\emptyset$ is a linear lapse rate (°C km$^{-1}$), and $E_g$ and $E_c$ (m) are the elevations of the temperature gage and the grid cell where $T_a$ is being calculated, respectively. Precipitation accumulates as SWE (m) when $T_a \leq T_{px}$, where $T_{px}$ is the freezing point (0°C by default). The precipitation $P$ is multiplied by a uniform multiplication factor ($S_{cf}$), which crudely represents snowpack sublimation and redistribution of snow due to wind (Anderson, 2006). The
resultant effective precipitation ($P_{eff}$) is added to the SWE.

Before the snowpack begins to melt, its heat deficit (or cold content) must be overcome. The change in heat deficit $\Delta D_t$ (mm of SWE), due to a temperature difference between the snow surface and air, is calculated as:

$$\Delta D_t = N_{mf,\max}(dt/6)\big(M_f/M_{f,\max}\big)(A_{TI} - T_{sur}), \tag{2}$$

where $T_{sur}$ is the snow surface temperature, and $A_{TI}$ is the antecedent temperature index (°C), which is calculated using $T_a$ and the antecedent snow temperature index parameter $A_{TIPM}$ (see Anderson (2006) for details regarding $T_{sur}$ and $A_{TI}$). $N_{mf,\max}$ is the maximum negative melt factor (mm °C$^{-1}$ (6 h)$^{-1}$), which is a parameter. $M_f$ is the melt factor (mm °C$^{-1}$ $dt^{-1}$), which is calculated as:

$$M_f = (dt/6)\big[S_v A_v\big(M_{f,\max} - M_{f,\min}\big) + M_{f,\min}\big], \tag{3}$$

where $S_v$ and $A_v$ are seasonal melt adjustments that change by Julian day, and $M_{f,\max}$ and $M_{f,\min}$ are the maximum and minimum melt factors (mm °C$^{-1}$ (6 h)$^{-1}$), which are parameters.

Once the heat deficit is overcome, SWE decreases as melt occurs. During normal conditions, the melt $M$ (mm of SWE) is:

$$M = \big[M_f(T_a - T_{mbase}) + 0.0125 P_{eff} f_r T_r\big]\Delta t, \tag{4}$$

where $T_{mbase}$ is the temperature at which melt begins (0°C by default), $f_r$ is the fraction of any precipitation that is rain (assumed equal to 1 when $T_a > 0°C$, otherwise set to 0), and $T_r$ is the precipitation temperature (assumed equal to $T_a$ or 0°C, whichever is greater). During rain-on-snow events (more than 1.5 mm of rainfall in the previous 6 h), $M$ is calculated from a simple energy balance:

$$M = \sigma\,[(T_a + 273)^4 - 273^4]\Delta t + 0.0125 P_{eff} f_r T_r + 8.5 f_u(\Delta t/6)[(r_h e_{sat} - 6.11) + 0.00057 P_a T_a], \tag{5}$$

where $\sigma$ is the Stefan-Boltzmann constant, $f_u$ is the average wind function (mm mb$^{-1}$ (6 h)$^{-1}$) (see Anderson (2006) for details), $r_h$ is the relative humidity (assumed to be 0.9 during rain-on-snow events) (Anderson, 1973, 2006), $P_a$ is atmospheric pressure (mb) (either measured or calculated from elevation) (Anderson, 2006), and $e_{sat}$ is the saturation vapor pressure (mb) (calculated based on Smith (1993)). The ripeness of the snowpack affects the amount of melt that is released and is controlled by the liquid holding capacity $L_{hc}$, which is a specified percentage of the ice in the snowpack (Anderson, 2006).

For frozen ground calculations, the snow depth is needed from the snow model. The snow depth $D_s$ (cm) is found from the SWE and the snowpack density. GSSHA uses the single-layer snow density functions from SNOW-17 (Anderson, 1976; Anderson, 2006). The density of newly fallen snow $\rho_n$ (gm cm$^{-3}$) varies between 0.05 ($T_a \leq -15°C$) and 0.15 ($T_a = 0°C$) according to:

$$\rho_n = 0.05 + 0.0017\,(T_a + 15)^{1.5}. \tag{6}$$

Increases in snowpack density $\rho_x$ from compaction, destructive metamorphism, and melt metamorphism due to the presence of liquid water are calculated as (Koren et al., 1999):

$$\rho_{x,t} = \rho_{x,t-1}\left(\frac{e^{B_2}}{B_2}\right)e^{B_1}, \tag{7}$$

where:

$$B_1 = c_3\, c_5\, dt\, e^{c_4\, T_s - c_x\, \beta\, (\rho_{x,t-1} - \rho_d)}, \text{ and} \tag{8}$$

$$B_2 = W_{t-1}\, c_1\, dt\, e^{0.08\, T_s - c_2 \rho_{x,t-1}}. \tag{9}$$

The variable $t$ is an index for time, $W$ is the ice portion of the snow pack (cm, $W = 100\, S_{swe,t-1}$) where $S_{swe}$ is the snow water equivalent on the ground in m, $T_s$ is the average snow pack temperature (°C, calculated based on Anderson (2006)), and $\rho_d$ is the threshold density above which destructive metamorphism decreases ($\rho_d$ is set to 0.15 gm cm$^{-1}$ based on Anderson (2006)). Finally, $\beta = 0$ if $\rho_{x,t-1} \leq \rho_d$, and $\beta = 1$ if $\rho_{x,t-1} > \rho_d$, and $c_1$ through $c_5$ are constants (see Anderson (2006) for details).

## 2.2 CFGI Frozen Ground Model

The CFGI model was originally developed as a lumped model for flood forecasting in the Pacific Northwest, but it has been used in distributed models as well (De Roo et al., 2001; Van Der Knijff et al., 2010). The rationale of the CFGI method is that air temperature ultimately controls the ground temperature, but its impact is moderated by the insulating effects of any snowpack. The presence of frozen ground is determined by the frozen ground index $F$ (°C-days), which is calculated as:

$$F_t = F_{t-1} A - T_{a,d}\, e^{-0.4 K_s D_s}, \tag{10}$$

where $T_{a,d}$ is the average daily air temperature (°C), $A$ is a daily decay coefficient, and $K_s$ is the snow reduction coefficient (cm$^{-1}$). $A$ controls the persistence of the $F$ values, and $K_s$ controls the insulation from the snowpack. Molnau and Bissell (1983) recommended changing $K_s$ depending on whether $T_{a,d}$ is above or below freezing (denoted as $K_{s,T_a>0°C}$ and $K_{s,T_a<0°C}$, respectively).

Higher values of $F$ indicate a higher likelihood that the ground is frozen. Once $F$ exceeds a specified threshold ($F_{threshold}$), the ground is considered frozen and infiltration is restricted. Molnau and Bissell (1983) found the ground to be frozen when $F > 83$ °C-days and thawed when $F < 56$ °C-days. When $F$ is between these values, the ground could be either frozen or thawed. It is worth noting that $F$ does not depend on soil moisture, which is known to affect the initialization and depth of frozen ground (Kurganova et al., 2007; Willis et al., 1961).

## 2.3 RTI Snowpack Model

The RTI model makes two modifications to the TI model: (1) it uses a radiation-derived temperature $T_{rad}$ (°C) to better describe the available energy, and (2) it estimates spatially-varying snowpack sublimation based on solar radiation approximations.

The RTI model replaces $T_a$ in Eq. (4) and (5) with a radiation-derived proxy temperature $T_{rad}$ (°C). In those equations, $T_a$ is used to conceptually represent the energy available to the snowpack. $T_{rad}$ has a similar purpose but is intended to improve the estimation of available energy. $T_{rad}$ is calculated by assuming that the radiation terms dominate the

energy balance at the snow surface so that outgoing longwave radiation balances the net incoming shortwave and longwave radiation (Follum et al., 2015). Thus:

$$R_{LW\uparrow} = R_{SW,net} + R_{LW\downarrow}, \tag{11}$$

where $R_{LW\uparrow}$ is outgoing longwave radiation, $R_{SW,net}$ is the net incoming shortwave radiation, and $R_{LW\downarrow}$ is the downwelling

longwave radiation. The right side of Eq. (11) represents the energy that is supplied to the snowpack via the atmosphere. $R_{LW\uparrow}$ (W m$^{-2}$) is the radiative response of the snowpack to that energy. Using the Stefan-Boltzmann Law, $R_{LW\uparrow}$ can be written in terms of a temperature $T_{rad}$:

$$T_{rad} = \left(\frac{R_{SW,net} + R_{LW\downarrow}}{\varepsilon_{snow}\,\sigma}\right)^{1/4} - 273.15, \tag{12}$$

where $\varepsilon_{snow}$ is the emissivity of snow (assumed to be 0.97) and $\sigma$ is the Stefan-Boltzmann constant.

$R_{SW,net}$ is calculated:

$$R_{SW,net} = (1 - \alpha_s)R_{SW\downarrow}, \tag{13}$$

where $\alpha_s$ is the albedo of the snowpack, which is calculated based on the time elapsed since the most recent snowfall and whether melt is occurring (Henneman and Stefan, 1999). $R_{SW\downarrow}$ is the incident shortwave radiation, which is calculated:

$$R_{SW\downarrow} = R_{SW,0}\,\varphi_r\,\varphi_{atm}\,\varphi_c\,\varphi_v\,\varphi_s\,\varphi_t, \tag{14}$$

where $R_{SW,0}$ is the solar constant (Liou, 2002), $\varphi_r$ accounts for distance from the Earth to the sun (based on Julian day (TVA, 1972)), $\varphi_{atm}$ accounts for atmospheric scattering (based on elevation (Allen et al., 2005)), $\varphi_c$ accounts for absorption by clouds (based on fractional cloud cover (TVA, 1972)), $\varphi_v$ accounts for vegetation (set equal to the vegetation transmission coefficient (Bras, 1990)), $\varphi_s$ accounts for the slope/aspect of the terrain (based on latitude, slope, and azimuth angle (Duffie and Beckman, 1980)), and $\varphi_t$ accounts for topographic shading (based on elevation, azimuth angle, and solar

elevation angle).

     $R_{LW\downarrow}$ is calculated from the contributions of the atmosphere (including clouds) and the canopy:

$$R_{LW\downarrow} = \sigma\varepsilon_a(T_a + 273.15)^4(1.0 + 0.17\,N^2)(1 - F_c) + F_c\sigma\varepsilon_c(T_{canopy} + 273.15)^4, \tag{15}$$

where $\varepsilon_a$ is the air emissivity (0.757 when snow is present based on Bras (1990)), $N$ is the fractional cloud cover, $F_c$ is the fractional canopy cover (estimated from leaf area index $L_{AI}$ following (Liston and Elder, 2006; Pomeroy et al., 2002)), $\varepsilon_c$ is

the canopy emissivity (assumed equal to 1 following Sicart et al. (2004)), and $T_{canopy}$ is the canopy temperature (°C) which is assumed equal to $T_a$ following DeWalle and Rango (2008).

     Because the TI model uses $T_a$ to drive snowpack dynamics, those dynamics are only directly associated with the downwelling longwave radiation from the air, which is a component of $R_{LW\downarrow}$. Furthermore, the spatial variations in the available energy depend only on the variations of $T_a$, which are inferred from elevation. $T_{rad}$ in the RTI model considers

both $R_{SW,net}$ and $R_{LW\downarrow}$ and thus accounts for heterogeneity in topographic orientation and shading as well as canopy cover. The TI model partially accounts for seasonal variation in solar radiation and snow albedo by empirically adjusting $M_f$ as

shown in Eq. (3). In the RTI model, seasonal variations in solar radiation and snow albedo are included in $T_{rad}$, so a constant melt factor $M_f$ is used (Follum et al., 2015).

The TI model uses a uniform multiplication factor ($S_{sf}$) that is applied to the precipitation to account for sublimation, but sublimation is known to vary spatially (Musselman, 2008; Rinehart, 2008; Veatch, 2009). Most sublimation methods depend on relative humidity and wind speed (e.g. Pomeroy, 1988; Liston and Elder, 2006), which are often unavailable in data sparse environments. However, Gustafson et al. (2010) linked differences in sublimation rates to the amount of solar radiation a location receives. In the RTI model a simple approach is used to estimate hourly sublimation rates $S_{sub}$(cm hr$^{-1}$) as:

$$S_{sub} = S_{sub,d} \left( \frac{R_{SW\downarrow}}{R_{SW\downarrow,flat}} \right),$$ (16)

where $S_{sub,d}$ (cm d$^{-1}$) is the watershed-average daily maximum sublimation amount (a parameter), and $R_{SW\downarrow,flat}$ is the daily shortwave radiation for a flat cell within the watershed on a cloud-free day. Thus, locations with higher $R_{SW\downarrow}$ (e.g., open areas and south-facing slopes in the northern hemisphere) will have higher values of $S_{sub}$. The method neglects wind speed and relative humidity, but does vary sublimation rates based on spatial patterns of solar radiation.

**2.4 modCFGI Frozen Ground Model**

The CFGI model is modified in three ways to create the modCFGI model. First, the average daily proxy temperature $T_{rad,d}$ is used in place of $T_{a,d}$ to represent available energy. Second, ground cover (leaf litter, woody debris, etc.) is included as an insulator in the frozen ground index. And third, an option is included to estimate frost depth based on the frozen ground index. The frost depth calculation is optional because it requires soil moisture estimates and may not be needed in many hydrologic models that only require the occurrence (not depth) of frozen ground.

The CFGI uses $T_{a,d}$ in Eq. (10) to represent the energy that is available to heat the ground surface. In the modCFGI model, $T_{a,d}$ is replaced with $T_{rad,d}$. $T_{rad,d}$ is calculated using $\alpha_s$ (see Eq. (12) and (13)) when snow is present, and the albedo of the land cover when snow is not present. By using $T_{rad,d}$, the modCFGI model is expected to better represent the spatial heterogeneity of energy supply due to variations in the topography and canopy cover within a watershed.

The insulation by the ground cover is included by modifying Eq. (10) to become:

$$F_t = F_{t-1}A - T_{rad}\, e^{-0.4(K_s D_s + K_{gc} D_{gc})},$$ (17)

where $K_{gc}$ is the ground cover reduction coefficient (cm$^{-1}$) and $D_{gc}$ is the depth of ground cover (cm). This formulation retains the original form of the CFGI model but includes insulation from both snowpack and ground cover. *F* can still be used to identify the occurrence of frozen ground, which may be sufficient for many hydrologic models. However, because *F* is not a measurable quantity, an option to extend modCFGI to calculate frost depth is also needed.

Frost depth is calculated using $F$ and the modified Berggren Equation. As originally proposed (and described by DeWalle and Rango (2008)), the Berggren equation relates the number of degree days in the freezing/thawing period $U$ (°C-days) to the maximum frost depth $Z_{max}$ (m) as follows:

$$Z_{max} = \lambda(48\ U\ \delta^{-1}\Omega_m)^{1/2}, \tag{18}$$

where $\lambda$ is a dimensionless coefficient that accounts for changes in sensible heat of the soil, $\delta$ (J m$^{-3}$) is the latent heat of fusion of the soil, and $\Omega_m$ (J m$^{-1}$ h$^{-1}$ °C$^{-1}$) is the mean thermal conductivity of the frozen and unfrozen soil layers. The derivation and corresponding assumptions (i.e. linear soil temperature gradients (Aldrich, 1956)) do not reveal any major impediments to adapting this equation for a shorter time step. In addition, Fox (1992), Woo et al. (2004), and Carey and Woo (2005) have used a layered version of the Stefan Equation, which is similar to Eq. (18) to simulate daily frost depths

with daily input data. Thus, the modified Berggren Equation is applied at a daily time scale and revised to become:

$$Z_d = \lambda[48\ (F - F_{threshold})\ \delta^{-1}\Omega_m]^{1/2}, \tag{19}$$

where $Z_d$ is the depth of frozen ground (m). By using the difference between $F$ and $F_{threshold}$, the degree-days of the current freezing/thawing period is utilized, which is similar to the use of $U$ in the original equation. $Z_d$ is only calculated once the ground begins to freeze (when $F > F_{threshold}$). $Z_d$ deepens as $F$ becomes increasingly larger than $F_{threshold}$.

When $F$ decreases (due to increasing $T_{rad}$), so does the thickness of frost depth. No frost remains when $F$ falls below $F_{threshold}$.

For the original modified Berggren Equation, $\lambda$ can be estimated annually from Aldrich (1956) using $U$, the mean annual air temperature, and the soil water content $\omega$ (% of dry weight). Here, $\lambda$ is calculated using daily differences between $F$ and $F_{threshold}$, the mean annual air temperature, and daily $\omega$ values. Thus, soil moisture is included in the calculation of

$Z_d$ even though it is not included in the calculation of $F$. Furthermore, $\delta$ is estimated daily as:

$$\delta = \delta_f \rho\ \omega/100, \tag{20}$$

where $\delta_f$ is the latent heat of fusion of water (0.334 MJ kg$^{-1}$ at 0°C) and $\rho$ is the dry soil density. $\Omega_m$ is estimated as (Farouki, 1981; Johansen, 1977):

$$\Omega_m = (\Omega_{sat} - \Omega_{dry})\omega + \Omega_{dry}, \tag{21}$$

where $\Omega_{dry}$ and $\Omega_{sat}$ are the thermal conductivity of dry and saturated soil, respectively. $\Omega_{sat}$ is calculated as the geometric mean of the conductivities of the materials within the soil profile (Farouki, 1981; Johansen, 1977):

$$\Omega_{sat} = \Omega_s^{(1-n_{total})}\Omega_{ice}^{(n_{ice})}\Omega_{water}^{(n_{total}-n_{ice})}, \tag{22}$$

where $\Omega_s$, $\Omega_{ice}$, and $\Omega_{water}$ are the thermal conductivity of solids, ice, and water, respectively (Farouki, 1981). $n_{total}$ is the porosity, and $n_{ice}$ is:

$$n_{ice} = n_{total}\ Z_d/H, \tag{23}$$

where $H$ (m) is the soil thickness.

## 3 Model Application

### 3.1 Study Area

The TI/CFGI and RTI/modCFGI models are tested at the W-3 sub-basin (Fig. 1) of the SREW. The study period is 1 Oct 2005 through 30 Sept 2010, which is water year (WY) 2006 through 2010. The SREW was founded in 1958 primarily for studies of snow accumulation, melt, and runoff (Anderson, 1973; Anderson, 1976; Dunne and Black, 1970a; Dunne and Black, 1970b; Dunne and Black, 1971; Shanley, 2000; Shanley and Chalmers, 1999). The W-3 sub-basin is located at 44° 29' N and 72° 09' W. Elevations range between 348 m and 697 m, and the area is approximately 8.5 km$^2$ (based on the National Elevation Dataset (Gesch et al., 2002)). The basin is primarily forested with deciduous (57.7%), evergreen (7.8%), and mixed (15.3%) trees (based on the 2006 National Land Cover Database (NLCD) (Fry et al., 2011)). Approximately 14.6% of the land cover is pasture/hay and cultivated crops. These open areas are typically below an elevation of 525 m, which is the approximate limit for viable agriculture (Shanley and Chalmers, 1999). The W-3 sub-basin is extensively gaged for both hydrometeorology and hydrology by the U.S. Geological Survey (USGS) and collaborators from federal agencies and universities. Additional basin information and data are provided by Shanley et al. (1995), Shanley and Chalmers (1999) and the USGS website (https://nh.water.usgs.gov/project/sleepers/index.htm, accessed 7 November 2016).

Two snow sites and 35 frost sites within W-3 were monitored by the Vermont Field Office of the USGS. At the snow sites, SWE and snow depth were measured approximately weekly, and both sites are used in the present study. At the frost sites, snow depth and frost depth were measured periodically (between 0 and 14 measurements in a given winter). Frost depth was measured using CRREL-Gandahl frost tubes (Ricard et al., 1976), which are filled with a methylene blue solution. The frost depth is identified by a change in colour within the tube (blue indicates thawed, clear indicates frozen). Vermette and Kanack (2012) provide images and descriptions of similar frost tubes, and Shanley and Chalmers (1999) provides detailed descriptions of the frost tubes at SREW. The frost sites (labelled FS in Fig. 1) are clustered in six parts of the watershed. For this paper, one site from each cluster (FS4, FS11, FS21, FS24, FS30, and FS40) was selected for analysis. The selected sites are far enough apart to be relatively independent but still capture the variations in elevation and land cover classification within the watershed.

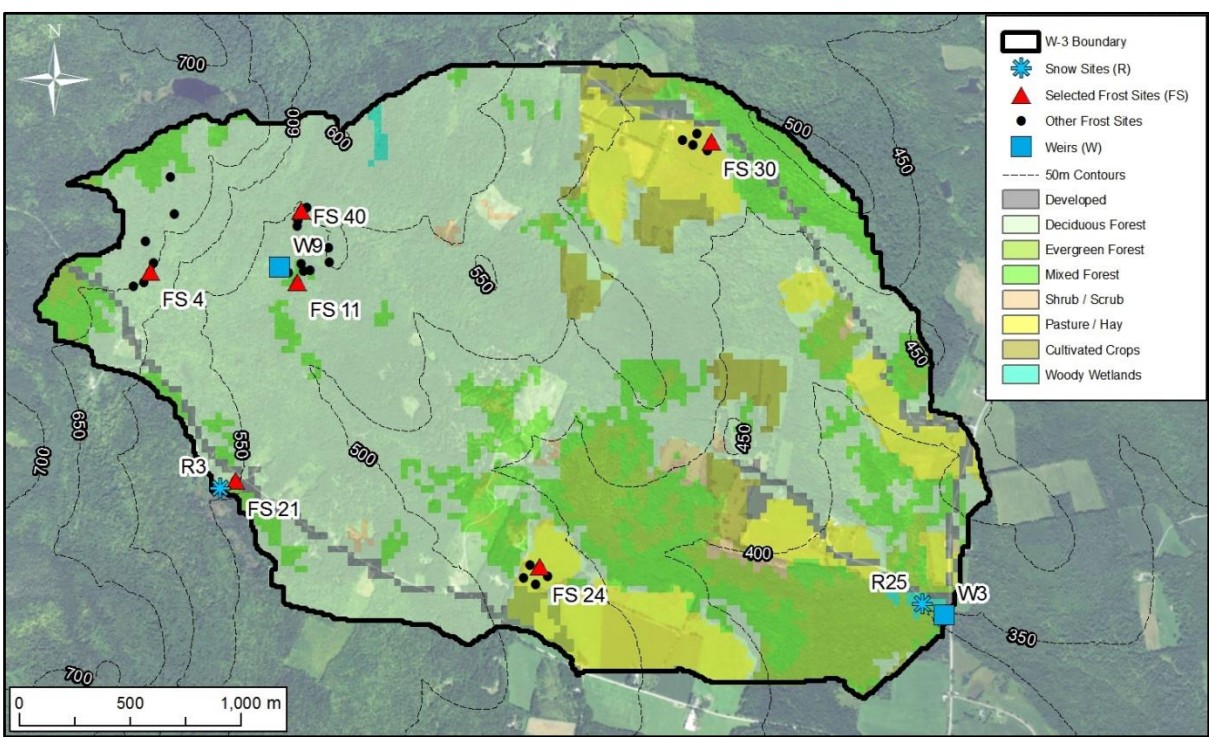

**Figure 1. W-3 sub-basin in the Sleepers River Experimental Watershed. Sites used in this study are identified with red triangles and blue snowflakes. Basin delineation and elevation contours (m) are based on the 1/3-arc-second National Elevation Dataset, land cover classification based on the 2006 National Land Cover Database, and sources of the background imagery include ESRI, DigitalGlobe, Earthstar Geographics, CNES/Airbus DS, GeoEye, USDA FSA, USGS, Getmapping, Aerogrid, IGN, IGP, and the GIS User Community.**

## 3.2 Model Inputs

The TI and CFGI models require hourly precipitation and temperature data, which were obtained from the USGS. Precipitation was measured at the W9 weir and R3 snow site (Fig. 1). The USGS then creates a single spatially-averaged precipitation time series by weighting the measurements using the distribution of elevation (based on personal communication with Dr. James Shanley of the Vermont Field Office of the USGS on 14 November 2016). The W9 gage receives more weight because the watershed includes elevations both above and below this site. Hourly temperature was measured at the W9 site, which has an elevation of 520 m.

The RTI and modCFGI models also require cloud cover data, which were obtained from the National Centers for Environmental Information (NCEI, https://www.ncdc.noaa.gov/, accessed 7 November 2016). The hourly cloud-cover classification data (clear, few clouds, broken sky, etc.) were collected at the Edward F. Knapp State Airport (44 km southwest of the basin) and the Morrisville-Stowe State Airport (36 km west of the basin). The classification data were converted to cloud cover percentages using the method from Follum et al. (2015). Cloud cover data are routinely measured

at most airports in the U.S. (data archived at NCEI) as well as many meteorological stations. For simulation of frost depth (and comparison to frost depth observations), soil moisture and evapotranspiration were also simulated. These two components additionally require hourly relative humidity, wind speed, and atmospheric pressure data, all of which were obtained from a meteorological station at the Fairbanks Museum in Saint Johnsbury, VT (11 km southeast of the basin) with
missing values filled-in using hourly data from the two airports.

      All the models require elevation data to determine the spatial patterns of snow and frozen ground. W-3 was delineated using the 1/3-arc-second (~9 m) National Elevation Dataset (Gesch et al., 2002). The RTI and CFGI models additionally require land cover classifications, which were obtained from the 2006 National Land Cover Database (Fry et al., 2011) and have a 30-m resolution. The classifications of some grid cells were changed to match the land covers observed in
the field. In particular, the grid cell containing R3 was changed from deciduous forest to pasture/hay, FS11 was changed from mixed forest to evergreen forest, and FS21 was changed from developed to mixed forest. Both FS24 and FS30 are classified as pasture/hay, where FS24 is a managed pasture and FS30 is an unmanaged pasture (personal communication with Ann Chalmers of the Vermont Field Office of the USGS on 15 November 2016). For example, during field observations in November 2016, FS24 had manure spread throughout the field, while FS30 was not fertilized.

Soil classification data are also required if calculating frost depth, which were obtained from the Digital General Soil Map of the Unites States (Soil Survey Staff, Natural Resources Conservation Service, United States Department of Agriculture, Web Soil Survey, available online at http://websoilsurvey.nrcs.usda.gov/. Accessed 10 August 2016). Almost the entire W-3 basin is classified as fine sandy loam. The Watershed Modeling System (Aquaveo, 2013) was used to develop the GSSHA model with a 30-m structured grid. This resolution is adequate to capture the spatial heterogeneity of
the basin while remaining computationally efficient.

## 3.3 Parameter Estimation and Calibration

      The Model-Independent Parameter Estimation and Uncertainly Analysis (PEST) method (Doherty et al., 1994) was used to calibrate 7 parameters in the TI model and 8 parameters in the RTI model. PEST is a nonlinear local search parameter estimator that calibrates numerous parameters simultaneously to produce the best fit between simulated results
and observations. WY 2006 and 2007 were used as the calibration period. The TI and RTI snow models were calibrated first to minimize the sum of the squared residuals between simulated and observed snow depths at the 8 sites (6 frost sites and 2 snow sites).

      Table 1 displays the allowable range, calibrated value, and sensitivity ranking for the calibrated snow parameters. Goodness of fit statistics as well as description of affects each parameter has on the snow simulations are described in the
Results and Discussion section. The allowable ranges for $A_{TIPM}$, $f_u$, $L_{hc}$, $N_{mf,\max}$, $M_f$, $M_{f,\max}$, and $M_{f,\min}$ are based on physical limitations and typical ranges in the literature (Follum et al., 2015). $L_{AI}$ can be estimated from seasonal and annual relationships to remotely-sensed normalized difference vegetation index (NDVI) values (Wang et al., 2005). However, snowpack affects the measurement of greenness in high latitude regions (Beck et al., 2006). Thus, $L_{AI}$ and $K_v$ values were

calibrated based on land cover classifications with forested land covers being categorized as deciduous forest (including deciduous forest, woody wetlands, and mixed forest) or evergreen forest. $L_{AI}$ and $K_v$ values for non-forested land cover classifications were set to 0.0 and 1.0, respectively. $T_{px}$ and $T_{mbase}$ were not calibrated (both are 0°C) because the temperature data were post-processed by the Vermont USGS and are expected to be accurate. By comparing the temperature

5 measurements at W9 and the Fairbanks Museum (elevation of ~212.4 m), $\emptyset$ was estimated at 6.6 °C km$^{-1}$. All snow density parameters are set based on Anderson (1973) and Anderson (2006).

The PEST results indicate that the TI model's snow depths are most sensitive to $S_{cf}$, $M_{f,\max}$, $A_{TIPM}$, and $M_{f,\min}$. For the RTI model, snow depths are most sensitive to $K_v$ (deciduous), $A_{TIPM}$, $L_{AI}$ (evergreen), and $M_f$ (Table 1). The calibrated deciduous $K_v$ is near the top of the allowable range (1.0) and $L_{AI}$ is near the bottom (0.103), indicating that the

10 snow in the deciduous forest behaves similarly to the open pasture areas where $K_v$=1 and $L_{AI}$=0.

Table 1. Allowable ranges and calibrated values for the TI and RTI model parameters using PEST. Dashes indicate parameters that are not required in the associated model. The sensitivity ranking for each parameter is shown in parentheses.

| Parameter | Units | Allowable Range | Calibrated Values TI | RTI |
|---|---|---|---|---|
| $M_{f,\max}$ | mm °C$^{-1}$ (6 h)$^{-1}$ | 0.001-2.400 | 1.017 (2) | -- |
| $M_{f,\min}$ | mm °C$^{-1}$ (6 h)$^{-1}$ | 0.001-1.600 | 0.001 (4) | -- |
| $M_f$ | mm °C$^{-1}$ (6 h)$^{-1}$ | 0.001-2.400 | -- | 0.391 (4) |
| $S_{cf}$ | fraction | 0.800-1.000 | 0.869 (1) | -- |
| $S_{sub,d}$ | cm d$^{-1}$ | 0.001-0.100 | -- | 0.068 (6) |
| $N_{mf,\max}$ | mm °C$^{-1}$ (6 h)$^{-1}$ | 0.001-2.400 | 0.002 (6) | 0.256 (8) |
| $f_u$ | mm mb$^{-1}$ (6 h)$^{-1}$ | 0.001-1.000 | 0.500 (7) | 0.500 (10) |
| $A_{TIPM}$ | fraction | 0.001-1.000 | 1.000 (3) | 0.992 (2) |
| $L_{hc}$ | fraction | 0.001-0.100 | 0.001 (5) | 0.001 (9) |
| $K_{v,deciduous}$ | fraction | 0.200-1.000 | -- | 0.969 (1) |
| $K_{v,evergreen}$ | fraction | 0.200-0.800 | -- | 0.308 (7) |
| $L_{AI,deciduous}$ | m$^2$ m$^{-2}$ | 0.100-1.000 | -- | 0.103 (5) |
| $L_{AI,evergreen}$ | m$^2$ m$^{-2}$ | 1.000-4.000 | -- | 1.000 (3) |

The CFGI and modCFGI frozen ground models were calibrated to minimize the sum of squared residuals between the simulated and observed frost depths at the 6 frost sites. For purposes of comparison the modified Berggren equation was also added to the CFGI model to calculate frost depth. Table 2 displays the allowable range, calibrated value, and sensitivity ranking of each calibrated frozen ground parameter. Goodness of fit statistics as well as description of affects each

parameter has on the frost depth simulations are described in the Results and Discussion section. $F_{threshold}$ was calibrated for both the CFGI and modCFGI models with the upper range based on Molnau and Bissell (1983). Three $K_{gc}$ values were calibrated for the modCFGI frozen ground model: one for the managed pasture site FS24 ($K_{gc,FS24}$), one for the unmanaged pasture site FS30 ($K_{gc,FS30}$), and one for all other frozen ground sites ($K_{gc}$).

Following Molnau and Bissell (1983), multiple combinations of $A$ (0.8 and 0.97), and $K_{s,T_a<0°C}$ and $K_{s,T_a>0°C}$ (0.08, 0.2, and 0.5) values were tested with $A = 0.97$, $K_{s,T_a<0°C} = 0.08$, and $K_{s,T_a>0°C} = 0.5$ producing frost indices that best replicate the rise and fall of the frost depth as well as the timing of the peak frost depth. Depth of ground cover for each land cover type was obtained from field observations in November 2016. Specifically, $D_{gc} = 6$ cm for deciduous forest (fallen leaves), $D_{gc} = 2$ cm for evergreen forest (fallen leaves), $D_{gc} = 4$ cm for pasture (grass), and $D_{gc} = 0$ cm for all other land

cover types.

The modified Berggren Equation requires soil moisture, which can be simulated using several methods in GSSHA (Downer and Ogden, 2006). To facilitate extension of these results to other hydrologic models, the commonly-used single-layer Green and Ampt infiltration model (Green and Ampt, 1911) with soil moisture redistribution between rainfall events (Ogden and Saghafian, 1997) is utilized to calculate infiltration. Soil moisture is tracked using a simple bucket approach, accounting for infiltration, evapotranspiration, and groundwater recharge as described in Downer (2007). The soil layer thickness ($H$) is set to 0.5 m for both the soil moisture calculations and frost depth equations. Soil infiltration parameters are set based on published values for the W-3 soil type (Downer and Ogden, 2006; Rawls et al., 1982; Rawls and Brakensiek, 1985; Rawls et al., 1983) and are shown in Table 3. Evapotranspiration, which can reduce the soil moisture, is simulated using a Penman Monteith approach (Monteith, 1965; Monteith, 1981) with parameters estimated based on land cover

(Downer and Ogden, 2006). The dry soil density ($\rho = 1137$ kg m$^{-3}$) and dry soil thermal conductivity ($\Omega_{dry} = 792$ J m$^{-1}$ h$^{-1}$ °C$^{-1}$) are set based on measurements of fine sandy loam by Nikolaev et al. (2013).

For the CFGI model, the calibrated $F_{threshold}$ value (Table 2) is relatively close to the lower bound value of 56°C-days found in Molnau and Bissell (1983). For the modCFGI model, the calibrated $F_{threshold}$ value is at the lower bound. The $F_{threshold}$ value is expected to be lower for the modCFGI model than the CFGI model. The modCFGI model

incorporates the insulation by ground cover directly using $K_{gc}$ and $D_{gc}$, whereas the CFGI model can only account for those effects by adjusting the $F_{threshold}$ value. It is also worth noting that $K_{gc,FS30}$ has a very low value (minimum of allowable range), which suggests that insulation from grass in an unmanaged pasture is very small. This could be the result of snow falling within the grass of the unmanaged pasture, thus making any insulating contribution from the grass very small.

**Table 2. Allowable ranges and calibrated values for the CFGI and modCFGI model parameters using PEST. Dashes indicate parameters that are not required in the associated model. The sensitivity ranking for the modCFGI parameters are shown in parentheses.**

| Parameter | Units | Allowable Range | Calibrated Values CFGI | Calibrated Values modCFGI |
|-----------|-------|-----------------|------|---------|
| $F_{threshold}$ | °C - days | 5.00-83.00 | 52.55 | 5.00 (3) |
| $K_{gc}$ | cm | 0.001-1.000 | -- | 1.033 (1) |
| $K_{gc,FS24}$ | cm | 0.001-1.000 | -- | 1.887 (2) |
| $K_{gc,FS30}$ | cm | 0.001-1.000 | -- | 0.001 (4) |

**Table 3. Values of soil parameters used to calculate soil moisture in the single-layer Green and Ampt infiltration model.**

| Parameter | Units | Value |
|-----------|-------|-------|
| *saturated hydraulic conductivity* | cm h$^{-1}$ | 2.040 |
| *effective porosity* | cm$^3$ cm$^{-3}$ | 0.407 |
| *residual water content* | cm$^3$ cm$^{-3}$ | 0.038 |
| *field capacity* | cm$^3$ cm$^{-3}$ | 0.166 |
| *wilting point* | cm$^3$ cm$^{-3}$ | 0.075 |
| *capillary head* | cm | 8.570 |
| *pore distribution arithmetic mean* | cm cm$^{-1}$ | 0.466 |

## 4 Results and Discussion

### 4.1 Snow Depth and SWE (TI vs RTI)

Figure 2 shows maps of simulated snow depth on 23 February 2007 from the TI and RTI snow models. The spatial variability in the TI snowpack is entirely based on elevation (due to the inference of local air temperature from elevation). Higher elevations have deeper snowpack due to lower air temperatures. The RTI snowpack also varies with elevation but shows variation due to land cover as well. In particular, pasture areas have slightly shallower snowpack than surrounding areas due to higher sublimation rates and higher absorbed shortwave radiation. North-facing slopes also have more snow than south-facing slopes due to lower absorbed shortwave radiation. Although no maps of observed snow depth are available for comparison, large-scale distributions of snowpack are known to be controlled by elevation, land cover, and slope/aspect (Fassnacht et al., 2017; Jost et al., 2007), which is more consistent with the RTI model.

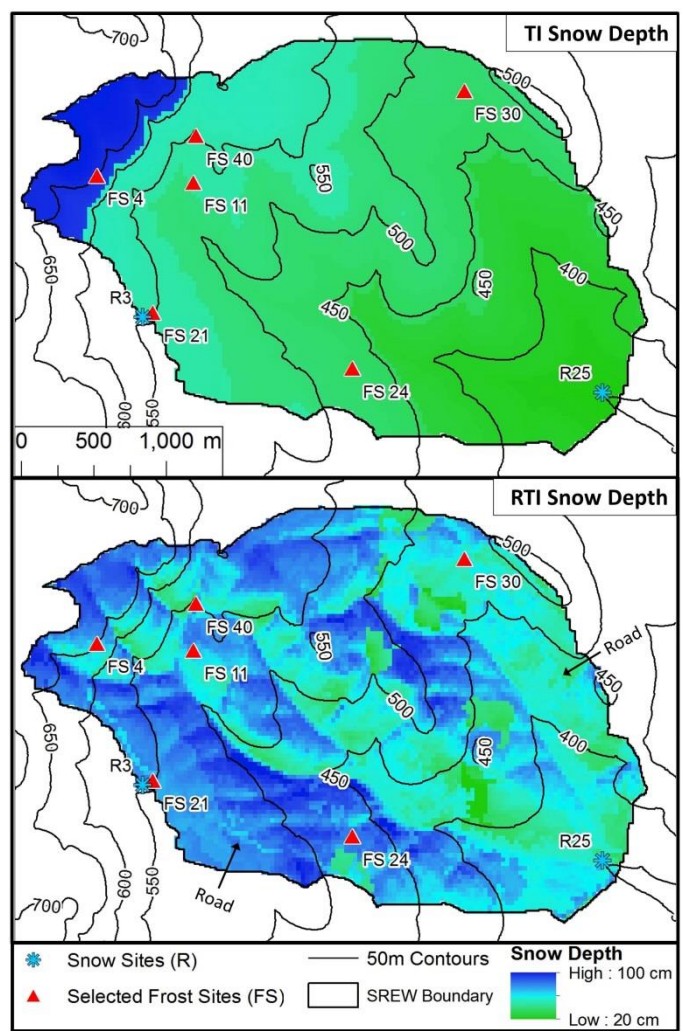

**Figure 2. Simulated maps of snow depth (TI and RTI models) within the W-3 watershed for 23 February 2007. No observed maps of snow depth are available, but the map shows the differences between the temperature-based (TI) model and the modified (RTI) model.**

Figure 3 shows the snow depths from the TI and RTI models at all 8 test locations and compares them to the observations. Root mean squared error (RMSE) and Nash-Sutcliffe Efficiency (NSE) are shown in Table 4 for the calibration period (WY 2006-2007), validation period (WY 2008-2010), and complete period (WY 2006-2010). The TI and RTI models track closely together at the 8 test locations despite differences in the snow depth shown in Fig. 2. Differences

10   between the TI and RTI snowpack at the test sites are small (Fig. 3 and Table 4). The RTI model performs slightly better than the TI model in overall average RMSE (15.69 vs. 15.71 cm), while the TI model performs slightly better in overall average NSE (0.58 vs. 0.53). The observed snow depth is relatively low in WY2008 and 2009 at two of the pasture sites (FS24 and FS30) compared to the other sites. Specifically in WY2008 the small snow depth observations are not captured

within either model. The R3 site is also classified as pasture yet has a higher snowpack in WY2008 and 2009. The higher snowpack at this pasture site may be explained by the proximity of R3 to forested areas, which may reduce the wind and help preserve the snowpack. Neither model considers wind effects.

The snow depths from the two models are similar at each location (Fig. 3) because on average the available energy to melt snow ($T_a$ in the TI model and $T_{rad}$ in the RTI model) is similar (Fig. 4). However, the diurnal variation of $T_{rad}$ is typically greater than that of $T_a$. $T_{rad}$ is derived from a simple radiation balance (i.e. neglecting other terms in the thermal energy balance). Thus, $T_{rad}$ is higher than $T_a$ during the day due to high $R_{SW\downarrow}$ values, and it is typically lower than $T_a$ at night because $R_{SW\downarrow}$ reduces to 0 and $\varepsilon_a$ (set to 0.757) in Eq. (15) limits the affect $T_a$ has on $R_{LW\downarrow}$ and therefore $T_{rad}$. As shown in Fig. 4, the available energy is also similar between these locations. The elevation difference between the highest

and lowest elevation site is approximately 300 m, corresponding to a maximum temperature difference of approximately 2°C between the sites. Also, the test sites are typically located on shallow slopes so topographic aspect has little influence on the energy available to melt the snowpack (i.e. $T_{rad}$). All land cover classifications except evergreen forest (FS11) have $K_v$ values at or near 1 and $L_{AI}$ values at or near 0, which reduces any variations due to land cover. $T_{rad}$ at FS11 (evergreen forest) is different from the other 7 sites because its low $K_v$ value (0.308) reduces $R_{SW,net}$ during the day, and a high $L_{AI}$

value (1.0) increases $R_{LW\downarrow}$ during day and night.

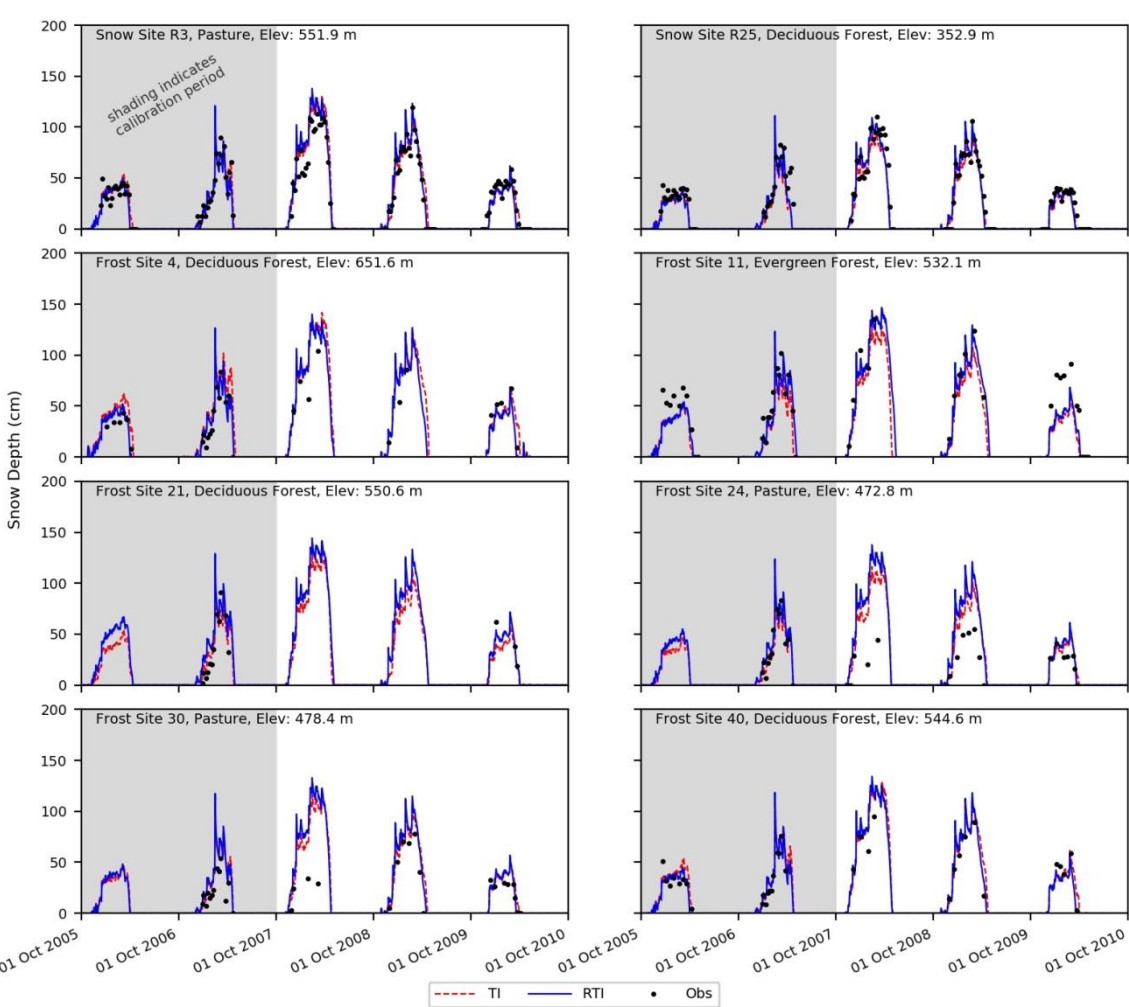

**Figure 3. TI and RTI simulated snow depth at all eight test sites within the W-3 watershed.**

**Table 4. Statistics for TI and RTI snow depth values at all 8 test sites, and statistics for TI and RTI SWE values at the R3 and R25 snow test sites. Values are shown for calibration period (WY 2006-2007), validation period (WY 2008-2010), and overall (WY 2006-2010). RMSE values closer to zero and NSE values closer to one indicate better fit.**

| | Site | Land Cover | Snow Model | Calibration RMSE (cm) | Calibration NSE | Validation RMSE (cm) | Validation NSE | Overall RMSE (cm) | Overall NSE |
|---|---|---|---|---|---|---|---|---|---|
| Snow Depth | R3 | Pasture | TI | 6.6 | 0.91 | 12.1 | 0.89 | 10.6 | 0.89 |
| | | | RTI | 8.4 | 0.86 | 11.9 | 0.89 | 10.9 | 0.89 |
| | R25 | Deciduous Forest | TI | 13 | 0.61 | 9.4 | 0.93 | 10.7 | 0.88 |
| | | | RTI | 12.2 | 0.65 | 9.2 | 0.93 | 10.3 | 0.89 |
| | FS4 | Deciduous Forest | TI | 18.1 | 0.3 | 17 | 0.57 | 17.7 | 0.51 |
| | | | RTI | 9.4 | 0.81 | 15.4 | 0.64 | 12.1 | 0.77 |
| | FS11 | Evergreen Forest | TI | 18.4 | 0.54 | 24.5 | 0.65 | 21.6 | 0.62 |
| | | | RTI | 15.8 | 0.66 | 19.3 | 0.78 | 17.6 | 0.75 |
| | FS21 | Deciduous Forest | TI | 12.1 | 0.82 | 13 | 0.45 | 12.3 | 0.79 |
| | | | RTI | 18.1 | 0.59 | 6.1 | 0.88 | 16.4 | 0.62 |
| | FS24 | Pasture | TI | 10.1 | 0.85 | 26.2 | -1.18 | 21.3 | 0.08 |
| | | | RTI | 10.7 | 0.83 | 33.2 | -2.51 | 26.7 | -0.44 |
| | FS30 | Pasture | TI | 16.1 | -0.12 | 22.6 | 0.09 | 20.3 | 0.06 |
| | | | RTI | 17 | -0.26 | 24.9 | -0.1 | 22.2 | -0.12 |
| | FS40 | Deciduous Forest | TI | 9.3 | 0.74 | 13.1 | 0.74 | 11.2 | 0.78 |
| | | | RTI | 7.6 | 0.83 | 11.1 | 0.81 | 9.3 | 0.85 |
| SWE | R3 | Pasture | TI | 2.1 | 0.9 | 3.3 | 0.91 | 3 | 0.91 |
| | | | RTI | 3.6 | 0.71 | 3.1 | 0.92 | 3.3 | 0.89 |
| | R25 | Deciduous Forest | TI | 5.2 | 0.29 | 2.7 | 0.92 | 3.7 | 0.83 |
| | | | RTI | 5 | 0.35 | 3.1 | 0.91 | 3.8 | 0.82 |

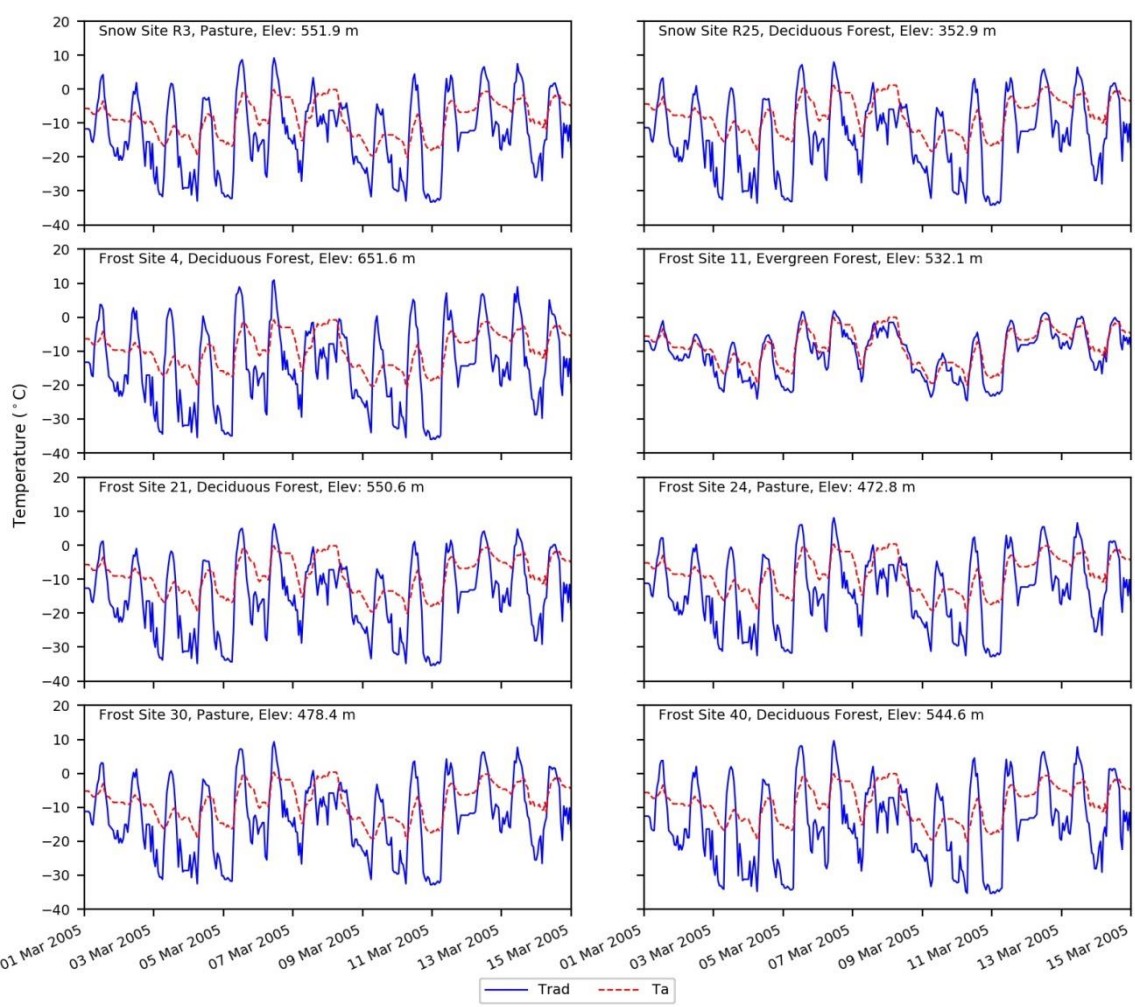

**Figure 4.** $T_a$ and $T_{rad}$ values at all eight test sites within the W-3 watershed between **01 March 2005 and 15 March 2005.**

Figure 5 shows the simulated (both TI and RTI models) and observed SWE values, and Table 4 shows the associated performance metrics at the R3 and R25 snow sites. The TI and RTI models are only calibrated to snow depth, but SWE is calculated first and then combined with snow density to determine snow depth. Both models use the same method to calculate snow density. Both models exhibit similar behaviour and performance at the two sites, which is consistent with their similar snow depths discussed earlier (Fig. 3 and Table 4). Overall, these suggest that the snow density equations used within GSSHA are relatively accurate at the W-3 watershed. Thus, accurate estimates of snow depth typically correspond to accurate estimates of SWE as well.

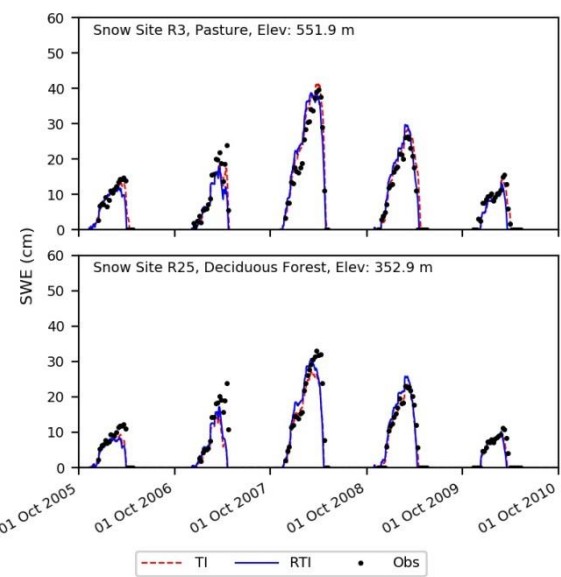

**Figure 5. TI and RTI simulated SWE at R3 and R25 snow sites within the W-3 watershed.**

## 4.2 Frost Depth (CFGI vs modCFGI)

Figure 6 shows simulated frost depth maps for 23 February 2007 using the CFGI and modCFGI models (no maps of observed frost depths are available for comparison). In the CFGI model, the frost depths mainly depend on elevation. Colder temperatures at higher elevations generally result in greater snowpack, which insulates the ground and produces smaller frost depths. However, at the beginning of the snow season when the snowpack is shallow, low temperatures at high elevations create deep frost in the higher elevations of the watershed. Later, deeper snowpack at high elevations insulate the

ground, while the frost depth increases at lower elevations. This reversal in the elevation dependence can produce an inversion (localized minima in frost depth), as seen between the 500 and 650 m contour lines in Fig. 6. The modCFGI frost depth also has some elevation dependence, but the spatial variation mainly follows land cover classification, which is similar to observations of frozen ground in the Swiss pre-alpine zone (Stähli, 2017). This variation is partly due to the use of $T_{rad}$ and the increased heterogeneity in the snow depth. The effect of snowpack can be seen by comparing hillslopes with the

same land cover but different orientations, such as along the 500 m contour south of FS11. Lower $T_{rad}$ values on northeast-facing slopes result in deeper snowpack than the southwest-facing slopes (Fig. 2). This deeper snowpack produces shallower frost depths on the northeast-facing slopes due to insulation by the snow. However, the spatial pattern of frost depth is more heavily affected by the land cover. Land cover's impact largely occurs through the associated ground cover. This effect can be seen by comparing the deep frost at the unmanaged pasture (near FS30) with the shallower frost depth at the deciduous

forest areas near FS4, FS21, and FS40. The low ground cover reduction coefficient at the unmanaged pasture ($K_{gc,FS30}$) reduces the insulation from the ground cover, creating deeper frost compared to the deciduous forest areas. The larger than

expected role of ground cover in the modCFGI model may occur because ground cover is present during the initiation, deepening, and decrease of frost depth, while the snowpack is much more variable throughout the season.

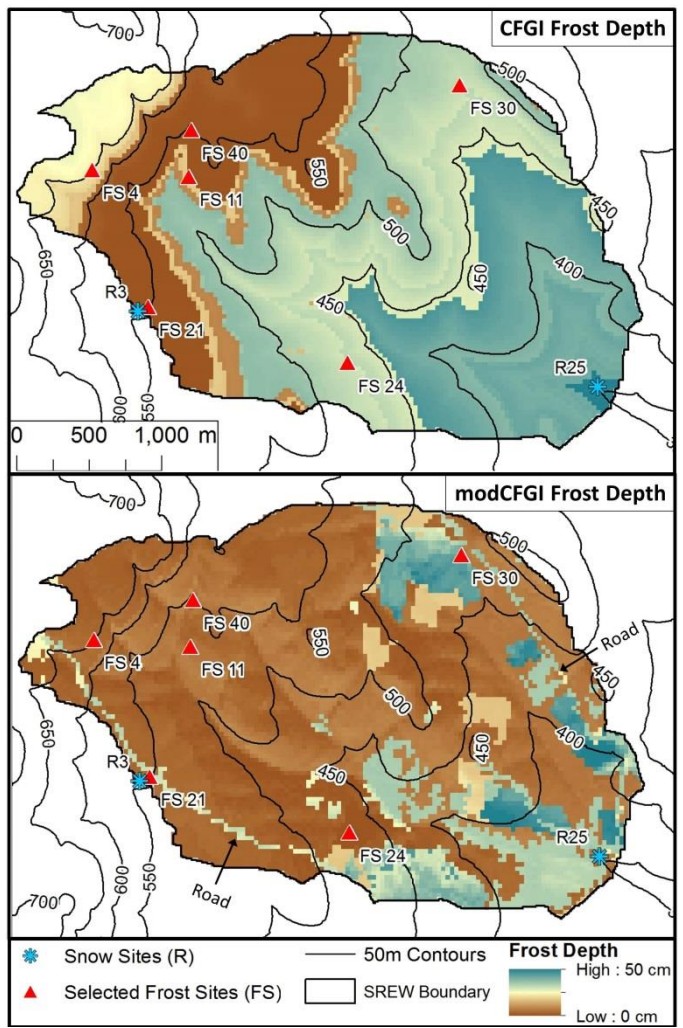

**Figure 6. Simulated maps of frost depth (CFGI and modCFGI models) within the W-3 watershed for 23 February 2007. No observed maps of frost depth are available, but the map shows the differences between the temperature-based (CFGI) model and the modified (modCFGI) model.**

Figure 7 shows the frost depths from the CFGI and modCFGI models along with the frost depth observations. The RMSE and NSE values during the calibration, validation, and overall periods are shown in Table 5. The simulated frost depth remains more constant amongst the sites when using the CFGI model, which produces similar maximum frost depths for a given year independent of the land cover. The modCFGI results deviate considerably from the CFGI results, producing greater frost depths at the unmanaged pasture (FS30) and evergreen (FS11) sites and smaller frost depths at the deciduous

(FS4, FS21, and FS40) and managed pasture (FS24) sites. These simulated differences between the sites are consistent with the observations. The decreased frost depth in the deciduous forest and managed pasture result from their high measured litter depth ($D_{gc} = 6$ cm) and high reduction coefficient ($K_{gc,FS24} = 1.887$ cm$^{-1}$), respectively. The two pasture sites (FS24 and FS30) differ considerably in the observed frost depth with FS30 consistently having deeper frost. This difference likely occurs because FS24 is managed and FS30 is not. With the exception of the validation period at FS30, the modCFGI model performs better (lower RMSE and higher NSE values) than the CFGI model. The difference in performance is most pronounced at the deciduous sites (FS4, FS21, and FS40) where the average overall NSE value is -11.9 for the CFGI model and 0.20 for the modCFGI model.

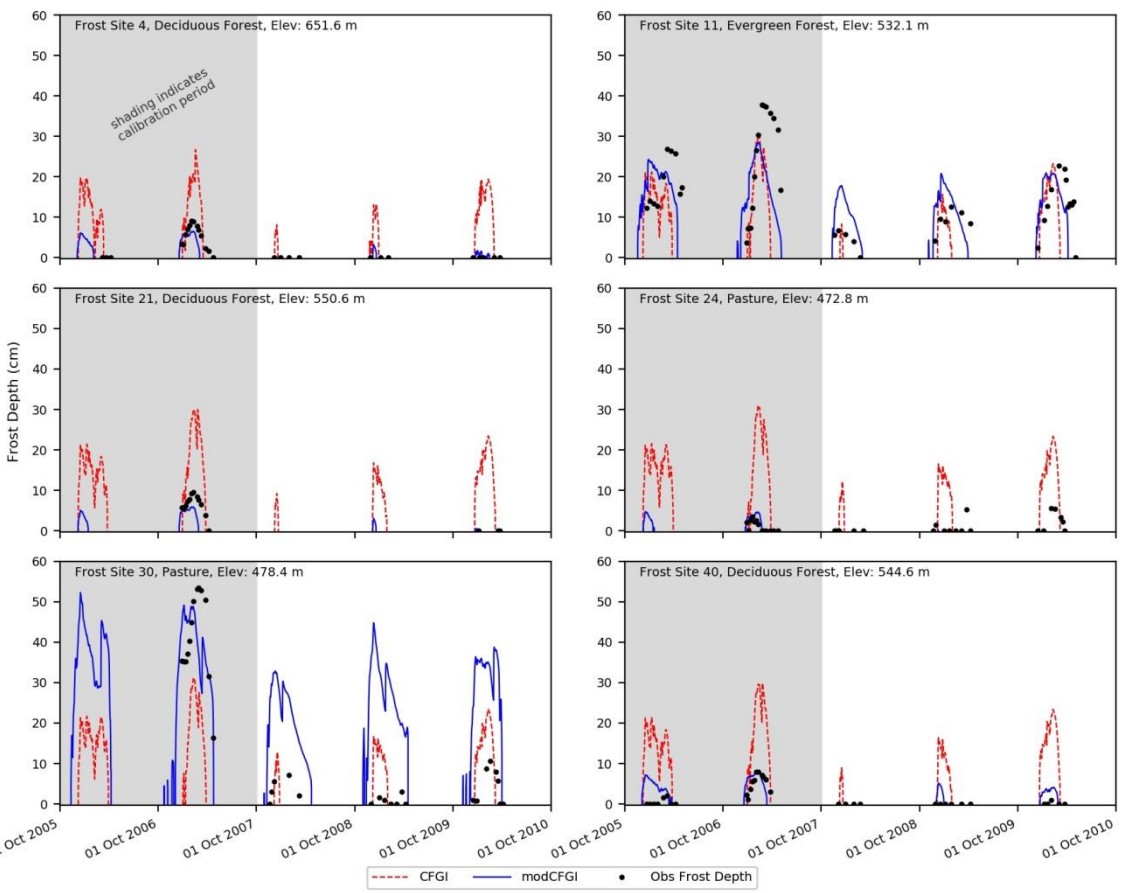

Figure 7. Observed frost depth compared against simulated (CFGI and modCFGI) frost depth at all 6 selected frozen ground test sites within the W-3 watershed.

**Table 5. Statistics for CFGI and modCFGI frost depth at all 6 frost sites. Values are shown for calibration period (WY 2006-2007), validation period (WY 2008-2010), and overall (WY 2006-2010). RMSE values closer to zero and NSE values closer to one indicate better fit. No frost was present at FS4 and FS21 during the validation period, resulting in an inability to calculate NSE. Statistics for a recalibrated modCFGI model without ground cover (labelled as "modCFGI no gc") are also shown.**

| Site | Land Cover | Frost Depth Model | Calibration | | Validation | | Overall | |
|---|---|---|---|---|---|---|---|---|
| | | | RMSE (cm) | NSE | RMSE (cm) | NSE | RMSE (cm) | NSE |
| FS4 | Deciduous Forest | CFGI | 8.2 | -5.0 | 5.7 | NA | 7.2 | -3.8 |
| | | modCFGI | 2.5 | 0.4 | 0.2 | NA | 1.9 | 0.7 |
| | | modCFGI no gc | 26.5 | -62.9 | 16.0 | NA | 22.6 | -45.9 |
| FS11 | Evergreen Forest | CFGI | 15.5 | -1.2 | 9.9 | -1.6 | 13.1 | -0.6 |
| | | modCFGI | 12.6 | -0.5 | 8.2 | -0.8 | 10.7 | -0.1 |
| | | modCFGI no gc | 17.5 | -1.8 | 10.9 | -2.1 | 14.6 | -1.0 |
| FS21 | Deciduous Forest | CFGI | 12.5 | -24.2 | 8.4 | NA | 11.8 | -10.9 |
| | | modCFGI | 3.9 | -1.5 | 0.0 | NA | 3.5 | -0.1 |
| | | modCFGI no gc | 26.2 | -109.8 | 14.3 | NA | 24.3 | -49.4 |
| FS24 | Pasture | CFGI | 17.5 | -188.5 | 7.3 | -12.3 | 12.4 | -49.1 |
| | | modCFGI | 1.4 | -0.3 | 2.3 | -0.3 | 2.0 | -0.3 |
| | | modCFGI no gc | 28.2 | -490.0 | 13.5 | -44.3 | 20.6 | -137.4 |
| FS30 | Pasture | CFGI | 27.5 | -5.8 | 6.2 | -2.4 | 17.7 | 0.2 |
| | | modCFGI | 11.7 | -0.2 | 24.9 | -55.0 | 20.9 | -0.1 |
| | | modCFGI no gc | 18.1 | -2.0 | 11.4 | -10.8 | 14.4 | 0.5 |
| FS40 | Deciduous Forest | CFGI | 14.2 | -22.6 | 10.1 | -1642 | 12.6 | -20.9 |
| | | modCFGI | 3.4 | -0.3 | 1.8 | -52.1 | 2.8 | -0.1 |
| | | modCFGI no gc | 36.9 | -157.9 | 21.8 | -7631 | 31.2 | -133.1 |

In hydrologic models, capturing the presence of frozen ground is important because even shallow frost with high moisture content (concrete frost) has the potential to impede infiltration (Dunne and Black, 1971). Therefore, the ability of the CFGI and modCFGI models to accurately capture the presence of frozen ground is evaluated. Whenever frost observations are available, the simulated frost depths are categorized as: True Positive (both simulated and observed data show frost), True Negative (both simulated and observed data show no frost), False Positive (simulated data shows frost but observed data shows no frost), or False Negative (simulated data shows no frost but observed data shows frost). Table 6 shows the number of observations in each category for each test site. The table also shows the model accuracy, which is calculated as the percent of the observations that are correctly classified (True Positive or True Negative). The CFGI and modCFGI models perform similarly in capturing True Positives at FS4, FS21, FS24, and FS40, while modCFGI has more True Positives at FS11 and FS30. The lower True Positives and higher False Negatives indicate that the CFGI model tends to underestimate the presence of frozen ground at FS11 and FS30. Overall, both the CFGI and modCFGI models capture most of the frozen ground events, with the modCFGI model performing better than the CFGI model at 5 sites and worse at 1

site (FS21). The average accuracy of the modCFGI model is 15.2% higher than the CFGI model, with the largest increase in accuracy at FS11 (29.8%).

**Table 6. Number of True Positive (both simulated and observed data show frost depth), True Negative (both simulated and observed data show no frost depth), False Positive (simulated data shows frost depth but observed data does not), and False Negative (simulated data shows no frost depth but observed data shows frost depth) occurrences during the entire test period. The Accuracy is the sum of the True Positive and True Negative divided by the total number of observations.**

| Site | Land Cover | Elevation (m) | Model | True Positive | True Negative | False Positive | False Negative | Accuracy (%) |
|------|-----------|---------------|-------|---------------|---------------|----------------|----------------|--------------|
| FS4 | Deciduous Forest | 651.6 | CFGI | 9 | 12 | 4 | 3 | 75.0% |
| | | | modCFGI | 9 | 15 | 1 | 3 | 85.7% |
| FS11 | Evergreen Forest | 532.1 | CFGI | 24 | 2 | 0 | 21 | 55.3% |
| | | | modCFGI | 39 | 1 | 1 | 6 | 85.1% |
| FS21 | Deciduous Forest | 550.6 | CFGI | 10 | 3 | 1 | 1 | 86.7% |
| | | | modCFGI | 8 | 4 | 0 | 3 | 80.0% |
| FS24 | Pasture | 472.8 | CFGI | 7 | 12 | 9 | 5 | 57.6% |
| | | | modCFGI | 6 | 20 | 1 | 6 | 78.8% |
| FS30 | Pasture | 478.4 | CFGI | 16 | 8 | 0 | 10 | 70.6% |
| | | | modCFGI | 26 | 1 | 7 | 0 | 79.4% |
| FS40 | Deciduous Forest | 544.6 | CFGI | 13 | 9 | 11 | 1 | 64.7% |
| | | | modCFGI | 13 | 12 | 8 | 1 | 73.5% |
| Total | | | CFGI | 79 | 46 | 25 | 41 | 65.4% |
| | | | modCFGI | 101 | 53 | 18 | 19 | 80.6% |

A simple test is employed to explore the modification that contributes most to the increased accuracy of the modCFGI model. This test removes ground cover from the modCFGI model, recalibrates, and then compares the results to observations. When ground cover is removed, the calibrated $F_{threshold}$ value is 83 °C-days, which is at the top of the calibration range. This change indicates that ground cover has a large impact on the appropriate value of this threshold. Figure 8 shows the simulated frost depths using the modCFGI model with and without ground cover for each test site. Performance metrics for the modCFGI model with and without ground cover are shown in Table 5. Variability in frost depth between the sites is diminished when ground cover is removed, leading to large errors between simulated and observed frost depth. When ground cover is removed, the frost depth results decrease in accuracy (higher RMSE values and lower NSE values) compared to the complete modCFGI model. The only exception is the overall period at FS30, which is also the only site where the CFGI model outperforms the full modCFGI model. These results suggest that inclusion of ground cover is an important reason why the modCFGI model outperforms the CFGI model.

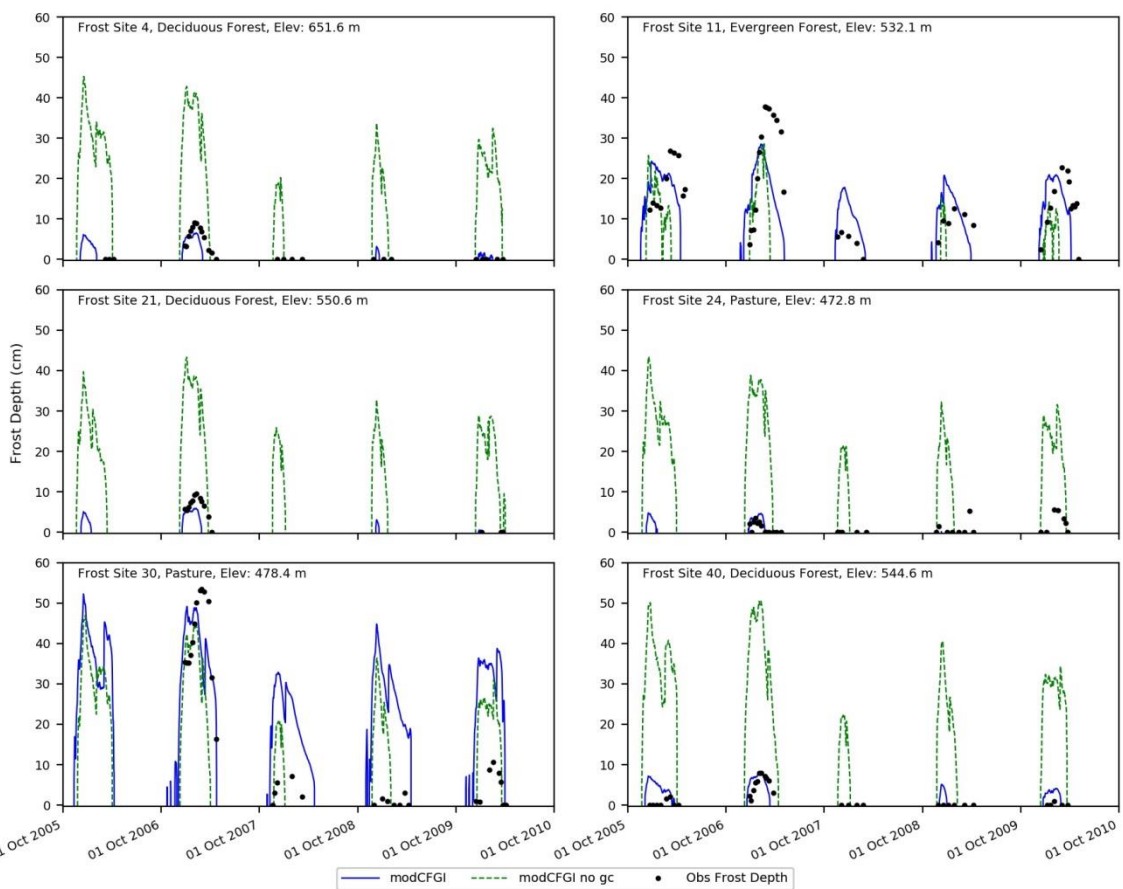

**Figure 8. Observed frost depth compared against simulated (modCFGI with and without ground cover included) frost depth at all 6 selected frozen ground test sites within the W-3 watershed. The modCFGI model without ground cover is labelled as "modCFGI no gc".**

The sensitivity of the modCFGI results to soil moisture is also examined. Soil moisture does not affect the calculation of $F$, but it is included within the modified Berggren Equation (Eq. (18) and (19)) in the calculation of $\delta$ (Eq. 20) and $\Omega_m$ (Eq. 21). Soil moisture was simulated using a single layer Green and Ampt approach. However, no soil moisture measurements are available at any of the test sites to evaluate the accuracy of the simulated values. Sensitivity of the modCFGI model to volumetric soil moisture is tested by artificially setting the soil moisture to either the residual water content ($\theta_{low}$) or the effective porosity ($\theta_{high}$), which are the lower and upper bounds for soil moisture values within the model. Figure 9 shows the modelled frost depths from the modCFGI model using $\theta_{low}$, $\theta_{high}$, and the soil moisture from the Green and Ampt approach ($\theta_{sim}$, which is identical to modCFGI in Fig. 7 and Fig. 8). Also shown are the observed frost depths for reference only. The frost depth from the $\theta_{sim}$ case is similar to the frost depth from the $\theta_{high}$ because the simulated soil moisture is usually close to the effective porosity. Frost depth increases when $\theta_{low}$ is used, which coincides with other studies (Fox, 1992; Willis et al., 1961). The timing of the frozen ground (when it begins and ends) is identical in

all three of the simulations. The consistent timing occurs because soil moisture is not used to calculate $F$ and the same $F_{threshold}$ (which controls when frozen ground begins) was used for all three simulations. This result highlights a deficiency in the modelling framework. Specifically, soil moisture should be considered for determining the initiation of frozen ground because wet soils have a higher specific heat capacity and require more energy loss to cool and freeze the soil (Kurganova et al., 2007).

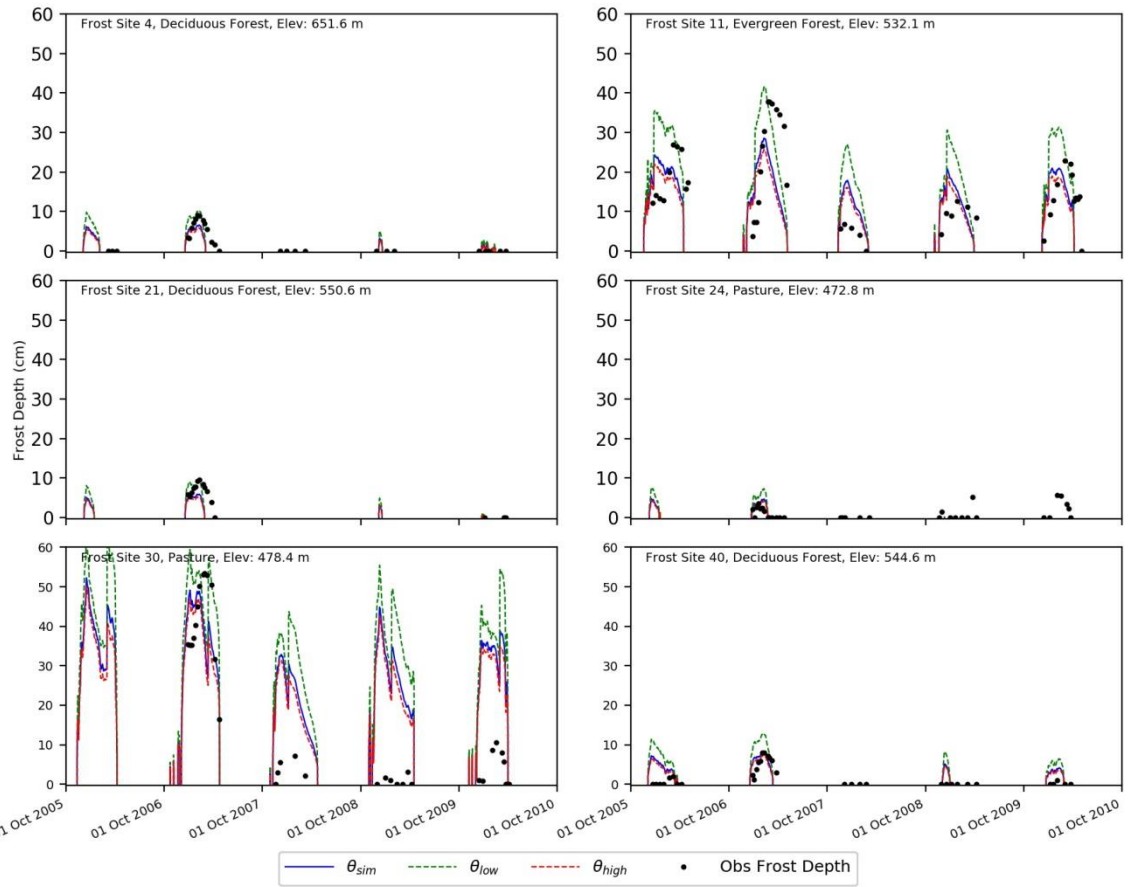

**Figure 9. Simulated frost depths from the modCFGI model using simulated soil moisture ($\theta_{sim}$), a constant high soil moisture ($\theta_{high}$), and a constant low soil moisture ($\theta_{low}$) at all 6 selected frozen ground test sites within the W-3 watershed.**

## 5 Conclusions

The main purpose of this paper was to better estimate the spatial pattern of frozen ground for distributed watershed modelling by modifying an existing degree-day frozen ground model (CFGI), which uses a frost index value to determine whether the ground is frozen or not. The modifications to the CFGI model include: 1) use of a radiation-derived temperature index (RTI) snow model instead of a standard temperature-index (TI) snow model, 2) use of a radiation-derived proxy temperature ($T_{rad}$) instead of air temperature ($T_a$) in the calculation of the frost index, 3) inclusion of ground cover (litter,

debris, grass, etc.) as an insulator of the ground from air temperatures, and 4) an option to use a version of the modified Berggren Equation to calculate frost depths based on the frost index values.  The CFGI and modCFGI models were tested using the GSSHA hydrologic model over a five-year period within the W-3 watershed, which is part of Sleepers River Experimental Watershed in Vermont.  The model results were compared against snow depth at eight sites, snow water equivalent at two sites, and frost depth at six sites.  The primary conclusions of the paper are as follows:

1.) The RTI snow model produces much more complex spatial patterns of snow depth than the TI snow model for the W-3 watershed.  The TI model, which is based on SNOW-17 (Anderson, 2006), only produces spatial variation using elevation.  The RTI model accounts for elevation, hillslope orientation, canopy shading, and longwave radiation from the canopy through the use of the radiation-derived proxy temperature.  It also includes a simple sublimation method based on solar radiation.  Thus, its snow depths exhibit spatial heterogeneity based on elevation, slope/aspect, and land cover, all of which are known to affect the largescale distribution of observed snow depths (Fassnacht et al., 2017; Jost et al., 2007).

2.) Both the RTI model and TI model produce accurate results for the eight snow depth sites at W-3.  Two of the eight sites also measure snow water equivalent, where the RTI and TI model also show similarly accurate results.  The eight test sites have similar topographic attributes and primarily differ in their land covers, which include pasture, deciduous forest, and evergreen forest.  Because the leaves have typically fallen prior to snow accumulation, all but the evergreen site behave similarly in snow accumulation and ablation.

3.) The modCFGI frost model produces more complex spatial patterns of frost depth than the CFGI frost model for the W-3 watershed.  The CFGI model uses elevation to infer the spatial variation of air temperature.  It also uses the TI model for snow depth, which also depends on elevation.  Thus, the simulated frost depths at W-3 primarily reflect the watershed elevations.  In contrast, the modCFGI model uses the radiation-derived proxy temperature to infer the energy available to heat the ground and the RTI model to simulate snow depth.  Furthermore, it accounts for the insulating effects of ground cover (in addition to snowpack), which also depends on the land cover.  Thus, the frost depths simulated by the modCFGI model at W-3 depend on the local elevation, hillslope orientation, and land cover, all of which are known to affect the distribution of frozen ground (Fox, 1992; MacKinney, 1929; Wilcox et al., 1997; Willis et al., 1961).

4.) The modCFGI model produces more accurate frost depths than the CFGI for all but one of the six test sites in the W-3 watershed.  Overall, the modCFGI model more accurately captures the inter-annual variability in frost depth at a given site and variability of frost depth between sites.  Although both the CFGI and modCFGI capture the majority of frozen ground events observed, the modCFGI model has 15.2% better accuracy in capturing the presence of frozen ground, which is expected to be important for capturing runoff that is produced by frozen ground.

5.) A key reason for the difference in performance between the two frost models is that the modCFGI model includes the insulation of the ground by ground cover while the CFGI model does not.  When ground cover is removed from

the modCFGI model its results for W-3 are less accurate and the variability in simulated frost depth between the sites is limited. Ground cover is likely important in this watershed because it is relatively thick and is also present at all stages of the winter while snowpack is not.

Overall, the modCFGI model provides improved spatial representation of frozen ground while requiring only cloud cover estimates as additional forcing data (more forcing data may be required if soil moisture is simulated to obtain frost depth). Limited data requirements should make modCFGI well-suited for data sparse environments. Hydrologic models often need to account for the presence of frozen ground, which in data-sparse environments often means using simple degree-day approaches that typically vary frozen ground with elevation only (as was shown with the CFGI model). To calculate $T_{rad}$ the modCFGI model does require cloud cover data, which are collected operationally at most airports within the U.S. If soil moisture is explicitly simulated within the hydrologic model the modCFGI model can also be used with the modified Berggren Equation to simulate frost depth, which requires information on soil type and an estimate of the thermal conductivity of the soil.

Five main avenues are available for future research. First, the modCFGI model should be generalized to include the effects of wind (as it relates to the snowpack) and more completely consider the role of soil moisture. Soil moisture is not considered when calculating the frost index, so it does not impact the initiation or duration of frozen ground. This limitation results from using a degree-day approach and may be important in some cases (Kurganova et al., 2007; Willis et al., 1961). Second, the modCFGI model should be tested further. Additional testing should consider other areas where snow and frozen ground are known to affect runoff, such as the Upper Midwest region of the United States. Additional testing should also better characterize the insulation properties of ground cover under different management scenarios. Third, the calculation of $T_{rad}$ is simple and applicable in data-sparse environments, but other approaches for adjusting a temperature value based on topography and land cover are available (Fox, 1992; Kang, 2005; Webster et al., 2017) and could be further tested. Fourth, future research should also determine the effects of spatial heterogeneity of snow and frost depth on runoff and streamflow at both the local and watershed scales. Similar to Campbell et al. (2010), the RTI and modCFGI models could be used in data-sparse watersheds to investigate how changes in historic and future climate affect snow, frozen ground, and runoff. Finally, although this paper focuses on the simulation of frost depth in the context of watershed modelling, the methods described could also be used for agriculture, overland mobility modelling, and infrastructure where snow and frost depth are major concerns.

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
