# Peer review of "A Simple Temperature-Based Method to Estimate Heterogeneous Frozen Ground within a Distributed Watershed Model"

_Hydrology and Earth System Sciences, 2017_

## Referee Comment (RC1) · Anonymous Referee #1 · 27 Sep 2017

The aim of this study was to come up with a simple index-based soil frost distribution model for hydrological purpose that can capture spatial variabilities in snow depth and vegetation cover. To this end, the authors extended an old, well-established frost index model (CFGI) and combined it with a modified temperature-index snow model. The snow and soil frost models were tested against data of five winters from an experimental watershed in Vermont. The results show some improvements of the simulated snow and soil frost depths, but they also highlight that simple, index-based models have their limitations in representing the temporal and spatial distribution of snow and soil frost.

[Figure]

The distribution of snow and soil frost depth is indeed a complex problem for northern-latitude and high-altitude regions of the world – in particular where a vegetation cover is present. Research studies with the objective to improve the simulation of snow and soil frost depth for hydrological purposes have been numerous in the past twenty years ... not only in north America, but also in Europe. For example, a physically based model (http://www.coupmodel.com/) was developed in Sweden some twenty years ago, has been made available (open access) and has been used for snow and soil frost simulations in Nordic and alpine countries, as well as for sites in Greenland. This model calculates the combined heat and water balance of a soil profile and accounts for interception processes and the presence of a variable snow cover (quite similar to the snow model in this manuscript). So what's the advantage of using a simpler index-based model?

In former times, the main argument was that such index-based models only needed one or few input variables (typically air temperature), whereas physically-based models need five or more meteorological input variables (cf. page 2, lines 11 and 12) and maybe a considerable amount of parameter settings. That's certainly a valid point for many applications. But here in this work, it seems that the new snow and soil frost model is not really simple and index-based, but is going in the direction of a more physically-based approach. Also it apparently needs not only air temperature, but also precipitation, wind speed, relative humidity and cloud coverage. Thus, the argument for developing a new index-based model (instead of using an established physically based model) is not obvious.

The snow and soil frost model modifications proposed in this work are reasonable. Improving a pure air temperature index snow model with a radiation term and accounting for snow interception is certainly of added value for an area such as the Sleepers River experimental watershed where the topography and the vegetation cover is distinctive. And adding a variable soil moisture content and a litter cover to the soil frost model is relevant as these are two important controls of soil frost formation. The core of the new models is a radiation-derived proxy temperature T(rad) instead of simple air temperature. It would be interesting to see somewhere in this manuscript how much T(rad) differs from T(air) in order to clarify whether or not this makes a big difference for the snow model compared to the other modifications. Speaking of T(rad) I can recommend the following very recent publication (EOS spotlight article https://eos.org/research-spotlights/how-the-micrometeorology-of-alpine-forests-affects-snowmelt) for reference: Webster, C., et al. 2017. J. Geophys. Res. Atmos.,122, doi:10.1002/2017JD026581.

The results presented in this manuscript confirm that the modifications made to the original index models are significant and reasonable. Nevertheless, Figures 6 to 8 also clearly show that – even with these more sophisticated models – it is still a long way to reproduce correctly year-to-year variabilities and spatial variabilities in soil frost depth. This is only partially due to the limitations of the model. On the other hand it also reflects the tremendous sensitivity and variability of soil frost in a real landscape. A nice example of such a complex variability in soil frost occurrence was presented in Journal of Hydrology, 546: 90-102, 2017.

The conclusions from this work are plausible, though not really exciting and surprising. And the proposed avenues for further research related to this topic are reasonable. Overall the manuscript is well written and easy to read.

Specific comments:

Page 2, line 17: "When the frost index exceeds a threshold, the soil is considered frozen and impermeable to infiltration." And Page 20, line 7: "...any frozen ground has the potential to impede infiltration and produce flooding" This is not really true. There is a bunch of literature showing that frozen soils are not always impermeable (depending on the ice content), and it needs a certain frost depth for the soil to become impermeable. This could be mentioned somewhere.

Page 8, line 30: "Frost depth was measured using gages described by Ricard et al.

[Figure]

(1976) and Shanley and Chalmers (1999)." Please provide some information what principle these measurements are based on. Are these frost tubes containing a blue liquid? Just referring to Ricard and Shanley is not good enough.

Pages 18 and 19: The results in Figures 5 and 6 show very nicely that frost depth is primarily controlled by the snow depth, which in turn is controlled by the forest canopy (or soil management). The air temperature decreasing with altitude is obviously of minor (or no) importance for this elevation range. This is an important message and agrees very well with observations from the lower alps in Switzerland (Journal of Hydrology, 546: 90-102, 2017).

Pages 25 to 29 (References): the cited literature is almost exclusively from North America, but there is a wealth of publications on the observation and spatial modelling of snow and frozen soil from Europe (and other parts of the world) that the authors seem to be unaware of.

The following reference is very relevant to this one and could be mentioned somewhere in the discussion: Campbell, J.L., Ollinger, S.V., Flerchinger, G.N., Wicklein, H., Hayhoe, K., Bailey, A.S., 2010. Past and projected future changes in snowpack and soil frost at the Hubbard Brook Experimental Forest, New Hampshire, USA. Hydrol. Process. 24, 2465–2480, doi: 10.1002/hyp.7666.

---

## Referee Comment (RC2) · Anonymous Referee #2 · 2 Oct 2017

The paper by Follum and co-authors seeks to develop a new 'simple' temperature based method to estimate frozen ground in a hydrology model. First, the issue of simulating frozen ground is critical for watershed models, particularly in much of the world where frozen ground strongly influences the rate, timing and magnitude of hydrological fluxes. There is a long history of incorporating degree-day and other frozen ground methodologies into hydrological models as the authors state, but of course they are by their nature highly calibrated. The authors state that frozen ground models that are more physically based (such as SHAW) are highly calibrated and state there is a need

for temperature index models that incorporate more physical parameters (i.e. ground cover, radiation derived temperature indices, etc). I have no doubt that this is the case, but in this paper the authors present a highly parametrized and calibrated degree-day model. It works okay, but certainly not great. In fact, I think there is enough forcing data here to drive SHAW and/or other more physically based land-surface schemes with frozen ground. Cloud cover and other radiation parameters are rarely measured operationally, and the adjustments of the TI portion of the model rely heavily on empirical adjustment. While going down the road of complexity, the authors include some new process representation (interception/unloading), while neglecting sublimation and others. The frozen ground model is adjusted to better improve physical representation, but it does not represent an advancement of our understanding of frozen ground process as the method of simulation is largely empirical and parameters are not transferrable. The strength of simple models is their ease of use and simplicity - but here we have a simple model that gets more and more complex and requires parmaterization that truly limits its applicability and does not justify it use when compared to existing models in the literature. While I can see that developing a local or improved freezing model is important in forecasting and operations, I do not believe that this paper in advances our understanding of frozen ground processes at a fundamental level that would justify publication in HESS. Perhaps a more operational journal would be the appropriate venue for this work.

―――――――――――――――

---

## Referee Comment (RC3) · Anonymous Referee #3 · 10 Oct 2017

The stated objective of this paper is to better estimate the spatial pattern of frozen ground in a gridded watershed model by modifying a simple cumulative freezing or thawing degree-day approach. This is basically accomplished by (1) using an air temperature index modified by a spatially variable solar radiation index (slope, aspect, elevation) and spatially variable canopy cover to model the snowpack, (2) using shortwave radiation and canopy cover in the calculation of a frost index, (3) using the insulating effects of ground cover when calculating a frost index, and (4), computing frost depth from the frost index by using the "a modified" Berggren Equation. The calculated snow depths and frost depths are then compared to measured snow and frost depths for locations within the Sleepers River Watershed for 5 winters. Snow depths and frost depths were also calculated using the unmodified temperature and frost indices. The results indicate that the modifications implemented generally improved simulations of snow depth, frost depth, and frost occurrence over the 8 spatially diverse sites.

I found that the Introduction to this paper was unnecessarily complicated. Basically, the authors used an air-temperature index modified by a potential solar radiation index (as affected by slope, aspect, and elevation), canopy cover and ground cover. This modified temperature index was then used to drive a snowpack accounting scheme and a frost depth calculation based on cumulative freezing degree-days.

I found the "frozen ground index" and its relationship to the "modified Berggren Equation" unclear. Is the modified Berggren Equation used only after the frozen ground index exceeds a "threshold" value?

Also, equation (16) on page 6 is puzzling to me. There seems to be an Rnet term (net all-wave radiation) missing. It should read: $Rnet = Rsw,net + Rlw\downarrow, - Rlw\uparrow$. Equation (16) makes sense only if Rnet is assumed to be zero but that is not stated.

It is unclear from the text if sublimation from the snowpack is accounted for when there is no canopy cover. It appears that sublimation is only calculated from intercepted snow.

Page 13, table 3: The term "residual saturation" is ambiguous, particularly at the low value of 0.038. Is this referring to some "degree of saturation", ie, some volumetric soil moisture expressed as a fraction of the saturated volumetric moisture content?

Page 23, line 4. Should read "requires more energy **loss** to cool and freeze the soil"

Overall, the authors are to be commended for their sound modeling procedures that included several field test sites, distinct calibration versus validation time periods, the sophisticated parameter estimation techniques, and sensitivity analysis. By using a 30-meter grid for capturing spatial variability they were able to generate informative maps of snow depth, frost depth, and snow water equivalent. The use of RMSE and NSE values for each case, as well as plots of absolute simulated and observed values for each time period, were appreciated. The absolute errors were hard to perceive for some data sets simply because of the small size of the

graphs.  It does appear from the plots that there were cases with high absolute differences between simulated and observed.

I agree with the authors assessment of the model's strengths and weaknesses, and the need for improved representation of the effects of wind and soil moisture in achieving better accuracy.

---

## Author Comment (AC1) · 22 Oct 2017

We would like to thank Referee #1 for their comments and suggestions. We appreciate all their insights about the paper and hope our responses address their suggestions and facilitate further discussion.

Main Issue #1, The frozen ground index method proposed requires numerous forcing data, therefore why use it instead of an energy balance approach?: The main goal of this study is to develop a frozen ground method that can be used within a

variety of watershed models, and we believe that the new modCFGI model has two significant advantages within this context. First, many watershed models do not explicitly simulate soil moisture, which would be required to implement an energy balance method. The modCFGI method can also use soil moisture to simulate the depth of frozen ground (as presented in this study), but soil moisture is not required to simulate the presence/absence of frozen ground. Thus, the modCFGI method can still be used to identify frozen soil for runoff production purposes even when soil moisture is not simulated. Second, the modCFGI method still requires fewer types of forcing data and fewer parameters than energy balance models. For example, the attached figure compares the forcing data requirements for two energy balance models (COUP and SHAW) along with the data requirements for the pre-existing CFGI model and the new modCFGI model. The energy balance models both require radiation data that are not readily available for many watersheds. In contrast, modCFGI model requires cloud cover data, which are routinely measured at most airports (data archived in the U.S. at the National Centers for Environmental Information, https://www.ncdc.noaa.gov/) as well as many meteorological stations. Although it is difficult to determine the total number of parameters that are required by energy balance models, the modCFGI likely requires specification of many fewer parameters, which reduces the potential for equifinality. In the revised paper, we plan to include a clearer statement of the paper's goals in the Introduction section, a short discussion of data requirements in the Model Application section, and a discussion of the model's use in data-sparse environments in the Conclusions.

Main Issue #2, Results are OK, but not exciting.: We agree that simulating temporal variability of frost depth remains a difficult problem for this model and others. However, the results in this paper are still a valuable contribution for three reasons. First, although temperature index models are commonly used in practice, very few studies have tested such models against observed frost depths. This study provides a rare evaluation of the existing CFGI model. Second, the proposed modCFGI model performs much better than the CFGI model in capturing the spatial variability of frozen ground within the

watershed (Figures 5 and 6). Figure 6 shows that the modCFGI model better captures the different frost depths at the various sites in the watershed. Accurate representation of the spatial pattern of frozen ground is expected to be important in capturing its role in flood production. Regarding temporal variability, the modCFGI better reproduces the high (e.g., WY 2007) and low (e.g. WY2008) frost depths and better captures the presence of frozen ground (Table 6). Third, the results suggest that litter depth is an important control on frost depth (last paragraph on page 21).

Specific Comment #1, Interesting to see comparison of radiation-derived proxy temperature (T_rad) to air temperature (T_a).: The values of T_rad tend to be higher than T_a when a cell has more direct sunlight (due to the cell's aspect and slope). Cloud and canopy cover also affect the T_rad pattern. In contrast, T_a in the CFGI model is estimated only using elevation. The impact of T_rad on snowpack was examined in Follum et al. (2015). We agree with the reviewer that a comparison between T_rad and T_a is important for the present study, and we will add this comparison to the results (likely in section 4.1) in the revised manuscript. Also, the effects of canopy that were highlighted in Webster et al. (2017) will be included in the discussion of the results.

Specific Edit Page 2, Line 17 "When the frost index exceeds a threshold, the soil is considered frozen and impermeable to infiltration.": We agree that frozen soils are not completely impermeable to infiltration, especially in forested environments (Lindstrom et al. 2002; Bayard et al., 2005; Nyberg et al., 2001; Shanley and Chalmers, 1999). However, this statement is referring to how some hydrologic models use degree-day frozen ground methods (such as CFGI) to restrict infiltration. In the revised manuscript, a clarification will be added (with citations) that indicates that this approach sometimes deviates from reality.

Specific Edit Page 20, Line 7 "…because any frozen ground has the potential to impede infiltration and produce flooding.": This statement will be modified to: "…because even shallow frost with high moisture content (concrete frost) has the potential to impede infiltration and produce flooding."

Specific Edit Page 8, Line 30 Description of frost tubes.: The CRREL-Gandahl frost tubes (Ricard et al., 1976) were used at SREW. The reviewer is correct—the frost tubes were filled with a methylene blue solution where freezing depth is identified by a change in colour within the tube (blue indicates thawed, clear indicates frozen). More details on the method of frost depth measurement will be provided on Page 8, with a reference to Vermette and Kanack (2012) who include images and descriptions of frost tubes that are similar to those used at SREW.

Specific Edit Pages 18 and 19, Figures 5 and 6: We agree with the reviewer that the change in elevation (and thus temperature) has small effect on snow and frozen ground within SREW. We appreciate the recommendation to cite the recent work by Stähli (2017), and we will include it in the results and discussion section.

Specific Edit Pages 25-29 Cited literature almost exclusively from North America.: We agree—research outside of North America will be added including numerical modelling approaches such as COUP (Jansson 2001; Jansson and Karlburg, 2010) and DWHC (Chen et al., 2007) and field investigations (Stähli 2017; Lindstrom et al. 2002; Bayard et al., 2005; Bayard and Stähli, 2005; Nyberg et al., 2001).

Specific Edit Pages 25-29 Inclusion of Campbell et al., (2010).: The research by Campbell et al. (2010) is related to our work and will be included in the Introduction and the Results and Discussion sections.

REFERENCES:

Chen, R.S., Kang, E.S., Ji, X.B., Yang, Y., Zhang, Z.H., Qing, W.W., Bai, S.Y, Wang, L.D., Kong, Q.Z, Lei, Y.H, Pei, Z.X.: Preliminary study of the hydrological processes in the alpine meadow and permafrost regions at the headwaters of Heihe River, Journal of Glaciology and Geocryology, 29(3), 387-396, 2007.

Bayard, D., Stähli, M.: Effects of frozen soil on the groundwater recharge in Alpine areas, Climate and hydrology in mountain areas. Wiley, Chichester, 73-83, 2005.

Bayard, D., Stähli, M., Parriaux, A., Flühler, H.: The influence of seasonally frozen soil on the snowmelt runoff at two Alpine sites in southern Switzerland, Journal of Hydrology 309(1), 66-84, 2005.

Flerchinger, G., Saxton, K.E.: Simultaneous heat and water model of a freezing snow-residue-soil system I, Theory and development, Transactions of the ASAE, 32(2), 565-0571, 1989.

Follum, M.L., Downer, C.W., Niemann, J.D., Roylance, S.M., Vuyovich, C.M.: A radiation-derived temperature-index snow routine for the GSSHA hydrologic model, Journal of Hydrology, 529, Part 3, 723-736, 2015.

Jansson, P.E.: Coupled heat and mass transfer model for soil-plant-atmosphere systems. 2001.

Jansson, P.E., Karlberg, L.: Coupled heat and mass transfer model for soil-plant-atmosphere systems, Royal Institute of Technology, Stockholm, p. 454, 2010.

Lindstrom, G., Bishop, K., Lofvenius, M.O.: Soil frost and runoff at Svartberget, northern Sweden—Measurements and model analysis, Hydrological Processes, 16, 3379–3392, 2002.

Molnau, M., Bissell, V.C.: A continuous frozen ground index for flood forecasting, Proceedings 51st Annual Meeting Western Snow Conference, Canadian Water Resources Association, Cambridge, Ontario, 109-119, 1983.

Nyberg, L., Stähli, M., Mellander, P.E., Bishop, K.: Soil frost effects on soil water and runoff dynamics along a boreal forest transact: 1. Field investigations, Hydrological Processes, 15, 909 – 926, 2001.

Ricard, J.A., Tobiasson, W., Greatorex, A.: The field assembled frost gage, United States Army Cold Regions Research and Engineering Laboratory, Hanover, New Hampshire, 1976.

[Figure]

Scherler, M. Hauck, C., Hoelzle, M., Stähli, M., Völksch, I.: Meltwater infiltration into the frozen active layer at an alpine permafrost site, Permafrost and Periglacial Processes 21(4), 325-334, 2010.

Shanley, J.B., Chalmers, A.: The effect of frozen soil on snowmelt runoff at Sleepers River, Vermont, Hydrological Processes, 13, 1843-1857, 1999.

Stähli, M.: Hydrological significance of soil frost for pre-alpine areas, Journal of Hydrology, 546, 90-102, 2017.

Vermette, S., Kanack, J.: Modeling frost line soil penetration using freezing degree-day rates, day length, and sun angle, 69th Eastern Snow Conference, Frost Valley YMCA, Claryville, New York, USA, 2012.

Webster, C., Rutter, N., Jonas, T.: Improving representation of canopy temperatures for modeling subcanopy incoming longwave radiation to the snow surface, Journal of Geophysical Research: Atmospheres 122(17), 9154-9172, 2017.

|  | Energy Balance | Energy Balance | Temperature-Index | Modified Temperature-Index |
|---|---|---|---|---|
|  | COUP Model | SHAW Model | CFGI Model[1] | modCFGI Model |
|  | (Scherler et al., 2011) | (Flerchinger and Saxton, 1989) | (Molnau and Bissel, 1983) | Proposed |
| Precipitation | ✓ | ✓ | ✓ | ✓ |
| Air Temperature | ✓ | ✓ | ✓ | ✓ |
| Relative Humidity | ✓ | ✓ |  | ✓ |
| Wind Speed | ✓ | ✓ |  |  |
| Global or Net Radiation | ✓ | ✓ |  |  |
| Incoming Long-Wave Radiation | ✓ |  |  |  |
| Cloud Cover |  |  |  | ✓ |

[1] Assumes CFGI is combined with a Temperature-Index Snow model (which requires precipitation)

[2] Using the RTI snow model which requires precipitation and relative humidity (both RTI and modCFGI require cloud cover)

**Fig. 1.** Required forcing data for the COUP, SHAW, CFGI, and modCFGI frozen ground models.

---

## Author Comment (AC2) · 24 Oct 2017

We would like to thank Referee #2 for their comments and suggestions. We appreciate all their insights about the paper and hope our responses address their suggestions and facilitate further discussion.

Main Issue #1, The frozen ground index method proposed is highly parameterized and requires many forcing data often not measured operationally (cloud cover and other radiation parameters).: The modCFGI method still requires fewer types of forcing data

and fewer parameters than energy balance models. For example, the attached figure compares the forcing data requirements for two energy balance models (COUP and SHAW) along with the data requirements for the pre-existing CFGI model and the new modCFGI model. The energy balance models both require radiation data that, as the reviewer mentions, are not readily available for many watersheds. In contrast, modCFGI model requires cloud cover data, which are routinely measured at most airports (data archived in the U.S. at the National Centers for Environmental Information, https://www.ncdc.noaa.gov/) as well as many meteorological stations. Although it is difficult to determine the total number of parameters that are required by the energy balance models, the modCFGI likely requires specification of many fewer parameters, which reduces the potential for equifinality. The modCFGI method can use soil moisture to simulate the depth of frozen ground (as presented in this study), but soil moisture is not required to simulate the presence/absence of frozen ground in this model. If only the presence/absence of frozen ground is required, the number of parameters further reduces. In the revised paper, we plan to include a short discussion of the data and parameter requirements in the Model Application section and a discussion of the model's use in data-sparse environments in the Conclusions.

Main Issue #2, New methods include some improved representation of the snowpack, but not all parts of the snowpack (e.g. sublimation).: The RTI snow model (see Section 2.3 on Page 5) maintains the same structure as the TI snow model, which is based on SNOW-17 (Anderson 1973; Anderson 2006). Like SNOW-17, both the RTI and TI models can account for interception / sublimation /condensation through an adjustable factor (SCF) (Anderson 2006; Follum et al., 2015), but this factor is typically applied uniformly to the watershed. Lines 19 and 20 on Page 3 will be modified to better reflect how the TI snow model accounts for interception and sublimation (via the SCF parameter). A clearer statement on page 5 will describe how in watersheds with multiple forest types (deciduous, evergreen, mixed, etc.) the interception, sublimation, and drip from the various canopies can be very different, and therefore a method (as applied in the RTI snow model) is needed to estimate these processes based on land cover type.

Main Issue #3, Modified model improves physical representation, but it does not represent an advancement of our understanding of frozen ground processes and is not transferable.: We believe this paper provides three significant advances in our understanding of frozen ground. First, it provides an evaluation of an existing temperature-index frozen ground model (the CFGI model). Although temperature index methods are often used in practice (lines 13-16, page 2), they have been rarely tested against observations of frost depth. Thus, the results provide useful insights into the performance of this class of models. Second, the new modCFGI model is better suited for use in a wide range of watershed models than other existing frozen ground models. Existing temperature index methods poorly reproduce the spatial variations in frozen ground because they do not fully account for the influence of topographic and canopy variations, as shown in this study (Conclusions 3-5 on page 24). Reproducing the spatial pattern of frozen ground is expected to be critical in capturing its role in flood production. The modCFGI model has better performance than the CFGI model in this respect (Figure 6 on page 19, and Table 6 on page 21). In comparison to energy balance models, the modCFGI model requires less forcing data and fewer parameters, and it does not require simulation of soil moisture (which is not explicitly simulated in many watershed models). Thus, we believe the new modCFGI model has significant practical value beyond its use in this study. Third, the study shows that much of the spatial variation of frozen ground in the watershed is controlled by insulating ground litter (Lines 5-7 on Page 2). We believe the role of litter cover has not been fully appreciated in previous studies. The method used to represent litter depth is also transferable to other models. In the revised paper, we plan to include a clearer statement of the paper's goals (in relation to the literature) in the Introduction section, a short discussion of data and parameter requirements in the Model Application section, and a discussion of the model's use in data-sparse environments in the Conclusions.

REFERENCES:

Anderson, E.A.: National Weather Service River Forecast System - Snow Accumulation and Ablation Model, Technical Memorandum NWS Hydro-17, November 1973, 217 pp., Silver Spring, Maryland, 1973.

Anderson, E.A.: Snow Accumulation and Ablation Model - SNOW-17, NWSRFS User Documentation, U.S. National Weather Service, Silver Springs, MD, 2006.

Bayard, D., Stähli, M.: Effects of frozen soil on the groundwater recharge in Alpine areas, Climate and hydrology in mountain areas. Wiley, Chichester, 73-83, 2005.

Flerchinger, G., Saxton, K.E.: Simultaneous heat and water model of a freezing snow-residue-soil system I, Theory and development, Transactions of the ASAE, 32(2), 565-0571, 1989.

Follum, M.L., Downer, C.W., Niemann, J.D., Roylance, S.M., Vuyovich, C.M.: A radiation-derived temperature-index snow routine for the GSSHA hydrologic model, Journal of Hydrology, 529, Part 3, 723-736, 2015.

Molnau, M., Bissell, V.C.: A continuous frozen ground index for flood forecasting, Proceedings 51st Annual Meeting Western Snow Conference, Canadian Water Resources Association, Cambridge, Ontario, 109-119, 1983.

Scherler, M. Hauck, C., Hoelzle, M., Stähli, M., Völksch, I.: Meltwater infiltration into the frozen active layer at an alpine permafrost site, Permafrost and Periglacial Processes 21(4), 325-334, 2010.

Shanley, J.B., Chalmers, A.: The effect of frozen soil on snowmelt runoff at Sleepers River, Vermont, Hydrological Processes, 13, 1843-1857, 1999.

[Figure]

| | Energy Balance
COUP Model
(Scherler et al., 2011) | Energy Balance
SHAW Model
(Flerchinger and Saxton, 1989) | Temperature-Index
CFGI Model[1]
(Molnau and Bissel, 1983) | Modified Temperature-Index
modCFGI Model
Proposed |
|---|---|---|---|---|
| Precipitation | ✓ | ✓ | ✓ | ✓ |
| Air Temperature | ✓ | ✓ | ✓ | ✓ |
| Relative Humidity | ✓ | ✓ | | ✓ |
| Wind Speed | ✓ | ✓ | | |
| Global or Net Radiation | ✓ | ✓ | | |
| Incoming Long-Wave Radiation | ✓ | | | |
| Cloud Cover | | | | ✓ |

[1] Assumes CFGI is combined with a Temperature-Index Snow model (which requires precipitation)

[2] Using the RTI snow model which requires precipitation and relative humidity (both RTI and modCFGI require cloud cover)

**Fig. 1.** Required forcing data for the COUP, SHAW, CFGI, and modCFGI frozen ground models.

---

## Author Comment (AC3) · 24 Oct 2017

We would like to thank Referee #3 for their comments and suggestions. We appreciate all their insights about the paper and hope our responses address their suggestions and facilitate further discussion.

Main Issue #1, Introduction unnecessarily complicated: In the revised manuscript we plan to simplify the introduction and better emphasize its goals in relation to the existing literature. The first, second, and third paragraphs in the Introduction will be condensed,

with more focus being placed on the objectives of this study (paragraphs four and five).

Main Issue #2, Relationship between frozen ground index and modified Berggren Equation is unclear.: The reviewer is correct that we use the modified Berggren Equation (as shown in Equation 22) to estimate frost depth once the ground begins to freeze, which in the CFGI and modCFGI models is when the frozen ground index F > F_Threshold. Frost depth continues to deepen as F values become increasingly larger than F_Threshold (Lines 21-23, Page 2). As the F values decrease (due to increased temperatures) so does the thickness of frost depth until no frost is left when F values fall below F_Threshold. In the revised manuscript, we plan to better describe the connection between F values calculated by CFGI / modCFGI and the modified Berggren Equation.

Main Issue #3, Net radiation term is missing from Equation 16.: Equation 16 represents a step in obtaining the radiation-derived proxy temperature (T_rad) for simulation of the snowpack. T_rad is calculated under the assumption that the outgoing long-wave radiation balances the net incoming short-wave and long-wave radiation (Follum et al., 2015). This balance is clearly an approximation, but the implied proxy temperature better represents the available energy than the air temperature. Follum et al. (2015) showed that the use of T_rad can provide a significant improvement in the simulation of snowpack in a temperature index approach. This assumption will be stated directly and discussed on Page 6.

Main Issue #4, Unclear on the inclusion of sublimation of the snowpack.: The RTI snow model (see Section 2.3 on Page 5) maintains the same structure as the TI snow model, which is based on SNOW-17 (Anderson 1973; Anderson 2006). Like SNOW-17, both the RTI and TI models can account for interception / sublimation /condensation through an adjustable factor (SCF) (Anderson 2006; Follum et al., 2015), but this factor is typically applied uniformly to the watershed. Lines 19 and 20 on Page 3 will be modified to better reflect how the TI snow model accounts for interception and sublimation (via the SCF parameter). A clearer statement on page 5 will describe how in watersheds with

multiple forest types (deciduous, evergreen, mixed, etc.) the interception, sublimation, and drip from the various canopies can be very different, and therefore a method (as applied in the RTI snow model) is needed to estimate these processes based on land cover type.

Main Issue #5, Ambiguous "residual saturation" term.: This term will be replaced with residual water content. In addition, we will clarify that all soil moisture parameters in Table 3 are volumetric quantities.

Minor Comment #1, Change in Line 4 of Page 23.: We appreciate the correction and will change the text from "require more energy to cool and freeze the soil" to "require more energy loss to cool and freeze the soil" on Line 4 of Page 23.

REFERENCES: Anderson, E.A.: National Weather Service River Forecast System - Snow Accumulation and Ablation Model, Technical Memorandum NWS Hydro-17, November 1973, 217 pp., Silver Spring, Maryland, 1973.

Anderson, E.A.: Snow Accumulation and Ablation Model - SNOW-17, NWSRFS User Documentation, U.S. National Weather Service, Silver Springs, MD, 2006.

Follum, M.L., Downer, C.W., Niemann, J.D., Roylance, S.M., Vuyovich, C.M.: A radiation-derived temperature-index snow routine for the GSSHA hydrologic model, Journal of Hydrology, 529, Part 3, 723-736, 2015.

---

## Author Response (AR1)

**A Simple Temperature-Based Method to Estimate Heterogeneous Frozen Ground within a Distributed Watershed Model**

Michael L. Follum[1,2], Jeffrey D. Niemann[2], Julie Parno[3], and Charles W. Downer[1]

[1]Coastal and Hydraulics Laboratory, Vicksburg, MS, 39180, USA.
[2]Department of Civil Engineering, Colorado State University, Fort Collins, CO, 80523, USA.
[3]Cold Regions Research and Engineering Laboratory, Hanover, NH, 03755, USA.

*Correspondence to*: Michael L. Follum (Michael.L.Follum@usace.army.mil)

**Summary of Author Response**

We would like to thank all three Referees for their comments and suggestions. We appreciate all of the insights about the manuscript that have led to several changes that we believe have made this a stronger submission. The response from the authors is broken into four sections followed by the "marked-up manuscript". The first three sections respond directly to comments made by each Referee, with the last section being a list of References. Each response includes the comments from the Referees, our response to the comment, and the manuscript changes that were made to address the comment. All line and page numbers refer to the "clean" manuscript (with no changes marked). The "marked-up manuscript" begins on Page 14.

**1 Referee #1**

**1.1 Comments from Referee #1**

The aim of this study was to come up with a simple index-based soil frost distribution model for hydrological purpose that can capture spatial variabilities in snow depth and vegetation cover. To this end, the authors extended an old, well-established frost index model (CFGI) and combined it with a modified temperature-index snow model. The snow and soil frost models were tested against data of five winters from an experimental watershed in Vermont. The results show some improvements of the simulated snow and soil frost depths, but they also highlight that simple, index-based models have their limitations in representing the temporal and spatial distribution of snow and soil frost. The distribution of snow and soil frost depth is indeed a complex problem for northern latitude and high-altitude regions of the world – in particular where a vegetation cover is present. Research studies with the objective to improve the simulation of snow and soil frost depth for hydrological purposes have been numerous in the past twenty years : : : not only in north America, but also in Europe. For example, a physically based model (http://www.coupmodel.com/) was developed in Sweden some twenty years ago, has been made available (open access) and has been used for snow and soil frost simulations in Nordic and alpine countries, as

well as for sites in Greenland. This model calculates the combined heat and water balance of a soil profile and accounts for interception processes and the presence of a variable snow cover (quite similar to the snow model in this manuscript). So what's the advantage of using a simpler index-based model?

In former times, the main argument was that such index-based models only needed one or few input variables (typically air temperature), whereas physically-based models need five or more meteorological input variables (cf. page 2, lines 11 and 12) and maybe a considerable amount of parameter settings. That's certainly a valid point for many applications. But here in this work, it seems that the new snow and soil frost model is not really simple and index-based, but is going in the direction of a more physically-based approach. Also it apparently needs not only air temperature, but also precipitation, wind speed, relative humidity and cloud coverage. Thus, the argument for developing a new index-based model (instead of using an established physically based model) is not obvious.

The snow and soil frost model modifications proposed in this work are reasonable. Improving a pure air temperature index snow model with a radiation term and accounting for snow interception is certainly of added value for an area such as the Sleepers River experimental watershed where the topography and the vegetation cover is distinctive. And adding a variable soil moisture content and a litter cover to the soil frost model is relevant as these are two important controls of soil frost formation. The core of the new models is a radiation-derived proxy temperature T(rad) instead of simple air temperature. It would be interesting to see somewhere in this manuscript how much T(rad) differs from T(air) in order to clarify whether or not this makes a big difference for the snow model compared to the other modifications. Speaking of T(rad) I can recommend the following very recent publication (EOS spotlight article https://eos.org/research-spotlights/how the-micrometeorology-of-alpine-forests affects- snowmelt) for reference: Webster, C., et al. 2017. J. Geophys. Res. Atmos., 122, doi:10.1002/2017JD026581.

The results presented in this manuscript confirm that the modifications made to the original index models are significant and reasonable. Nevertheless, Figures 6 to 8 also clearly show that – even with these more sophisticated models – it is still a long way to reproduce correctly year-to-year variabilities and spatial variabilities in soil frost depth. This is only partially due to the limitations of the model. On the other hand it also reflects the tremendous sensitivity and variability of soil frost in a real landscape. A nice example of such a complex variability in soil frost occurrence was presented in Journal of Hydrology, 546: 90-102, 2017.

The conclusions from this work are plausible, though not really exciting and surprising. And the proposed avenues for further research related to this topic are reasonable. Overall the manuscript is well written and easy to read.

Specific comments:

Page 2, line 17: "When the frost index exceeds a threshold, the soil is considered frozen and impermeable to infiltration."

And Page 20, line 7: ": : :any frozen ground has the potential to impede infiltration and produce flooding" This is not really true. There is a bunch of literature showing that frozen soils are not always impermeable (depending on the ice content), and it needs a certain frost depth for the soil to become impermeable. This could be mentioned somewhere.

Page 8, line 30: "Frost depth was measured using gages described by Ricard et al. (1976) and Shanley and Chalmers (1999)." Please provide some information what principle these measurements are based on. Are these frost tubes containing a blue liquid? Just referring to Ricard and Shanley is not good enough.

Pages 18 and 19: The results in Figures 5 and 6 show very nicely that frost depth is primarily controlled by the snow depth, which in turn is controlled by the forest canopy (or soil management). The air temperature decreasing with altitude is obviously of minor (or no) importance for this elevation range. This is an important message and agrees very well with observations from the lower alps in Switzerland (Journal of Hydrology, 546: 90-102, 2017).

Pages 25 to 29 (References): the cited literature is almost exclusively from North America, but there is a wealth of publications on the observation and spatial modelling of snow and frozen soil from Europe (and other parts of the world) that the authors seem to be unaware of.

The following reference is very relevant to this one and could be mentioned somewhere in the discussion: Campbell, J.L., Ollinger, S.V., Flerchinger, G.N., Wicklein, H., Hayhoe, K., Bailey, A.S., 2010. Past and projected future changes in snowpack and soil frost at the Hubbard Brook Experimental Forest, New Hampshire, USA. Hydrol. Process. 24, 2465–2480, doi: 10.1002/hyp.7666.

**1.2 Response to Referee #1**

Main Issue #1: The frozen ground index method proposed requires numerous forcing data, therefore why use it instead of an energy balance approach?

The main goal of this study is to develop a frozen ground method that improves spatial simulation of frozen ground and can be used within a variety of watershed models in data-sparse environments. This main goal is now more clearly stated in Lines 14-16 on Page 1 and Lines 30-32 on Page 2. We believe that the new modCFGI model has two significant advantages over energy balance approaches within this context. First, many watershed models do not explicitly simulate soil moisture, which would be required to implement an energy balance method (Lines 13-14 on Page 2). The modCFGI method has an option to use soil moisture to simulate the depth of frozen ground, but soil moisture is not required to simulate the presence/absence of frozen ground (Line 10 on Page 3; Lines 18-20 on Page 7; and Line 1 on Page 27). Thus, the modCFGI method can still be used to identify frozen soil for runoff production purposes even when soil moisture is not simulated. Second, the modCFGI method now requires fewer

types of forcing data than many energy balance models, making it more applicable in data sparse regions (Lines 5-7 on Page 28).

The RTI model in the initial manuscript used a common approach for estimating sublimation as described in Liston and Elder (2006), which requires precipitation, temperature, relative humidity, and wind speed data. To reduce the data requirements, a simpler sublimation method based on solar radiation approximations (already calculated in the RTI model) is now used (Lines 3-13 on Page 7). With this revision, the RTI/modCFGI approach requires precipitation, air temperature, and cloud cover forcing data. All of these variables are routinely measured at most airports (data archived in the U.S. at the National Centers for Environmental Information, https://www.ncdc.noaa.gov/) as well as many meteorological stations (Line 19 on Page 10 – Line 1 on Page 11; and Lines 9-11 on Page 28). If frost depth is required, then relative humidity, pressure, and wind speed forcing data may still be required to simulate evapotranspiration and soil moisture.

To better address the reviewer's comments, the revised paper now includes a clearer statement of the paper's goals (Lines 14-16 on Page 1; and Lines 30-32 on Page 2), a short discussion of data and parameter requirements in the Model Application section (starting on Line 8 on Page 10), and a discussion of the model's use in data-sparse environments in the Conclusions (Lines 5-13 on Page 28).

Main Issue #2: Results are OK, but not exciting.

We agree that simulating temporal variability of frost depth remains a difficult problem for this model and others. However, the results in this paper are still a valuable contribution for three reasons. First, although index-based models are commonly used in practice, very few studies have tested such models against observed frost depths. This study provides a rare evaluation of the existing CFGI model. Second, the proposed modCFGI model performs much better than the CFGI model in capturing the temporal and spatial variability of frozen ground within the watershed (Figures 6 and 7, Tables 5 and 6). Figure 7 and Table 5 show that the modCFGI model better captures the different frost depths at the various sites in the watershed. Regarding temporal variability, the modCFGI better reproduces the high (e.g., WY 2007) and low (e.g. WY2008) frost depths in Figure 7 and better captures the presence of frozen ground (Table 6). Third, the study shows that much of the spatial variation of frozen ground in the watershed is controlled by insulating ground litter (Lines 18-19 on Page 24; and Conclusion 5 on Page 27). Although ground cover is well known to affect frozen ground, we believe the role of litter cover has not been fully appreciated in previous studies using index-based approaches. The method used to represent litter depth is transferrable to other index-based frozen ground models.

Specific Comment #1: Interesting to see comparison of radiation-derived proxy temperature ($T_{rad}$) to air temperature ($T_a$).

Figure 4 on Page 19 was added to compare $T_{rad}$ and $T_a$ at all 8 test plots for a two week period in March 2005. The figure is also used to understand the similarities and differences between the snow simulations at the test plots (Figure 3), specifically related to the role of topography and vegetation on the energy balance (Lines 4-15 on Page 16). Because the plot sites are typically located along shallow slopes where terrain has limited influence, Figure 4 mostly highlights the differences in vegetation. The impact of $T_{rad}$ on snowpack was also examined in Follum et al. (2015), where four sites at approximately the same elevation but varying in aspect and vegetation type were examined and showed that topography has a dominant role in estimating $T_{rad}$ and therefore the snowpack. Because $T_{rad}$ relies on relationships between topography/vegetation and temperature, the work by Webster et al. (2017) was also included as a potential means to improve the estimation of $T_{rad}$ (Lines 20-22 on Page28).

Specific Edit Page 2, Line 17: Inaccurate Statement – "When the frost index exceeds a threshold, the soil is considered frozen and impermeable to infiltration."

We agree that frozen soils are not completely impermeable to infiltration, especially in forested environments (Lindstrom et al. 2002; Bayard et al., 2005; Nyberg et al., 2001; Shanley and Chalmers, 1999). However, this statement is referring to how some hydrologic models use index-based frozen ground methods (such as CFGI) to restrict infiltration. Lines 20-22 on Page 2 were added to clarify that this is often an incorrect assumption.

Specific Edit Page 20, Line 7: Inaccurate Statement – "…because any frozen ground has the potential to impede infiltration and produce flooding."

This statement was modified to: "…because even shallow frost with high moisture content (concrete frost) has the potential to impede infiltration" (Lines 6-7 on Page 23). A reference to Dunne and Black (1971) is also provided because they observed thin layers of concrete frost increasing the runoff production from plots within SREW.

Specific Edit Page 8, Line 30: Improve description of frost tubes.

The CRREL-Gandahl frost tubes (Ricard et al., 1976) were used at SREW. The reviewer is correct—the frost tubes were filled with a methylene blue solution where freezing depth is identified by a change in color within the tube (blue indicates thawed, clear indicates frozen). A better description of the frost tubes is now given in Lines 18-21 on Page 9. A reference to Vermette and Kanack (2012) is included because they provide images and descriptions of frost tubes that are similar to those used at SREW. A reference to Shanley and Chalmers (1999) is included because they describe the implentation of the frost tubes at SREW.

Specific Edit Reference recent literature in regards to Figure 5 & 6 on Pages 18 and 19:

We agree with the reviewer that the change in elevation (and thus temperature) has small effect on snow and frozen ground within SREW (now shown in greater detail with addition of Figures 4). We appreciate the recommendation

to cite the recent work by Stähli (2017), which is now referenced in Line 30 on Page 1; Lines 4 and 28 on Page 2; and Line 13 on Page 20.

Specific Edit Pages 25-29: Cited literature almost exclusively from North America

We agree. Research from outside North America is now better represented in this paper, especially in the first two paragraphs of the Introduction (starting on Line 25 of Page 1). Some of the added references include: Bayard et al. (2005); Bayard and Stähli (2005); Chen et al. (2007); Jansson (2001); Jansson and Karlburg (2010); Lindstrom et al. (2002); Nyberg et al. (2001); and Stähli (2017).

Specific Edit Pages 25-29: Include Campbell et al., (2010) as a reference.

The research by Campbell et al. (2010) is now included in the Introduction (Line 30 on Page 1) and Conclusions (Line 24, Page 28) sections.

**2 Referee #2**

**2.1 Comments from Referee #2**

The paper by Follum and co-authors seeks to develop a new 'simple' temperature based method to estimate frozen ground in a hydrology model. First, the issue of simulating frozen ground is critical for watershed models, particularly in much of the world where frozen ground strongly influences the rate, timing and magnitude of hydrological fluxes. There is a long history of incorporating degree-day and other frozen ground methodologies into hydrological models as the authors state, but of course they are by their nature highly calibrated. The authors state that frozen ground models that are more physically based (such as SHAW) are highly calibrated and state there is a need for temperature index models that incorporate more physical parameters (i.e. ground cover, radiation derived temperature indices, etc). I have no doubt that this is the case, but in this paper the authors present a highly parametrized and calibrated degree-day model. It works okay, but certainly not great. In fact, I think there is enough forcing data here to drive SHAW and/or other more physically based land-surface schemes with frozen ground. Cloud cover and other radiation parameters are rarely measured operationally, and the adjustments of the TI portion of the model rely heavily on empirical adjustment. While going down the road of complexity, the authors include some new process representation (interception/unloading), while neglecting sublimation and others. The frozen ground model is adjusted to better improve physical representation, but it does not represent an advancement of our understanding of frozen ground process as the method of simulation is largely empirical and parameters are not transferrable. The strength of simple models is their ease of use and simplicity - but here we have a simple model that gets more and more complex and requires parmaterization that truly limits its applicability and does not justify it use when compared to existing models in the literature. While I can see that developing a local or improved freezing model is important in forecasting and

operations, I do not believe that this paper in advances our understanding of frozen ground processes at a fundamental level that would justify publication in HESS. Perhaps a more operational journal would be the appropriate venue for this work.

**2.2 Response to Referee #2**

Main Issue #1, The frozen ground index method proposed is highly parameterized and requires many forcing data often not measured operationally (cloud cover and other radiation parameters).:

> The main goal of this study is to develop a frozen ground method that improves spatial simulation of frozen ground and can be used within a variety of watershed models in data-sparse environments. This main goal is now more clearly stated in Lines 14-16 on Page 1 and Lines 30-32 on Page 2. Although it is difficult to determine the total number of parameters that are required by energy balance models, the modCFGI likely requires specification of many fewer parameters, which reduces the potential for equifinality. The modCFGI method now also requires fewer types of forcing data than many energy balance models, making it more applicable in data sparse regions (Lines 5-7 on Page 28).

> The RTI model in the initial manuscript used a common approach for estimating sublimation as described in Liston and Elder (2006), which requires precipitation, temperature, relative humidity, and wind speed data. To reduce the data requirements, a simpler sublimation method based on solar radiation approximations (already calculated in RTI model) is now used (Lines 3-13 on Page 7). With this revision, the RTI/modCFGI approach requires precipitation, air temperature, and cloud cover forcing data. We agree with the reviewer that radiation data are often difficult to obtain operationally. However, operational precipitation, temperature, and cloud cover data are all routinely measured at most airports (data archived in the U.S. at the National Centers for Environmental Information, https://www.ncdc.noaa.gov/) as well as many meteorological stations (Line 19 on Page 10 – Line 1 on Page 11; and Lines 9-11 on Page 28). The modCFGI method also has an option to use soil moisture to simulate the depth of frozen ground, but soil moisture is not required to simulate the presence/absence of frozen ground (Line 10 on Page 3; Lines 18-20 on Page 7; and Line 1 on Page 27). Thus, the modCFGI method can still be used to identify frozen soil for runoff production purposes even when soil moisture is not simulated, providing a means to simulate frozen ground in hydrologic models that do not explicitly simulate soil moisture content (Line 14 on Page 2; and Lines 7-9 on Page 28). If soil moisture simulation is needed for frost depth estimation, the inclusion of relative humidity, pressure, and wind speed forcing data may still be required to simulate evapotranspiration and soil moisture.

> To better address the reviewer's comments, the revised paper now includes a clearer statement of the paper's goals (Lines 14-16 on Page 1; and Lines 30-32 on Page 2), a short discussion of data and parameter requirements in the

Model Application section (starting on Line 8 on Page 10), and a discussion of the model's use in data-sparse environments in the Conclusions (Lines 5-13 on Page 28).

Main Issue #2, New methods include some improved representation of the snowpack, but not all parts of the snowpack (e.g. sublimation).:

The reviewer is correct—neither the TI nor RTI snow models include all processes within the snowpack. The main purpose for using the RTI snow model is to better represent the spatial heterogeneity of the snowpack in data-sparse regions. Specifically, it includes melt from both short- and long-wave radiation through use of $T_{rad}$, which Follum et al. (2015) showed improves the spatial simulation of the snowpack by better accounting for topography and vegetation. We agree that some processes such as the movement of snow by wind are neglected by both the TI and RTI models.

Like SNOW-17, both the RTI and TI models can crudely account for interception / sublimation /condensation through a precipitation multiplication factor ($S_{cf}$) (Anderson 2006) typically applied uniformly to the watershed (Lines 28-30 on Page 3; and Lines 3-4 on Page 7). The initial manuscript set $S_{cf}$ equal to 1, while calculating canopy sublimation in the RTI model using the common approach described in Liston and Elder (2006). The updated manuscript calibrates $S_{cf}$ in the TI model and in the RTI model uses spatially-varying estimates of sublimation based on solar radiation. The simple method (described in Lines 3-13 on Page 7) estimates sublimation based on the incident solar radiation (and thus land cover and topography). The method neglects wind speed and relative humidity, which affect sublimation rates, but those variables are not available in many cold-region watersheds. Nonetheless, it does vary sublimation rates based on prevalence to solar radiation, which can control sublimation rates and therefore the variability of the snowpack (Gustafson et al., 2010).

Main Issue #3, Modified model improves physical representation, but it does not represent an advancement of our understanding of frozen ground processes and is not transferrable.:

We believe this paper provides three significant advances in our understanding of frozen ground. First, it provides an evaluation of an existing degree day frozen ground model (the CFGI model). Although index-based methods are often used in practice, they have been rarely tested against observations of frost depth. Thus, the results provide useful insights into the performance of this class of models. Second, the new modCFGI model is better suited for use in a wide range of watershed models than other existing frozen ground models. Existing index-based methods poorly reproduce the spatial variations in frozen ground because they do not fully account for the influence of topographic and canopy variations, as shown in this study (Conclusions 3-5 on Pages 27 and 28). Reproducing the spatial pattern of frozen ground is expected to be critical in capturing its role in flood production. The modCFGI model has better performance than the CFGI model in this respect (Figures 6 and 7, and Tables 5 and 6). In

comparison to energy balance models, the modCFGI model requires less forcing data and fewer parameters, and it does not require simulation of soil moisture (which is not explicitly simulated in many watershed models). Thus, we believe the new modCFGI model has significant practical value beyond its use in this study. Third, the study shows that much of the spatial variation of frozen ground in the watershed is controlled by insulating ground litter (Lines 18-19 on Page 24; and Conclusion 5 on Page 27). Although ground cover is well known to affect frozen ground, we believe the role of litter cover has not been fully appreciated in previous studies using index-based approaches. The method used to represent litter depth is transferrable to other index-based frozen ground models.

To better address the reviewer's comments, the revised paper now includes a clearer statement of the paper's goals (Lines 14-16 on Page 1; and Lines 30-32 on Page 2), a short discussion of data and parameter requirements in the Model Application section (starting on Line 8 on Page 10), and a discussion of the model's use in data-sparse environments in the Conclusions (Lines 5-13 on Page 28).

**3 Referee #3**

**3.1 Comments from Referee #3**

The stated objective of this paper is to better estimate the spatial pattern of frozen ground in a gridded watershed model by modifying a simple cumulative freezing or thawing degree-day approach. This is basically accomplished by (1) using an air temperature index modified by a spatially variable solar radiation index (slope, aspect, elevation) and spatially variable canopy cover to model the snowpack, (2) using shortwave radiation and canopy cover in the calculation of a frost index, (3) using the insulating effects of ground cover when calculating a frost index, and (4), computing frost depth from the frost index by using the "a modified" Berggren Equation. The calculated snow depths and frost depths are then compared to measured snow and frost depths for locations within the Sleepers River Watershed for 5 winters. Snow depths and frost depths were also calculated using the unmodified temperature and frost indices. The results indicate that the modifications implemented generally improved simulations of snow depth, frost depth, and frost occurrence over the 8 spatially diverse sites.

I found that the Introduction to this paper was unnecessarily complicated. Basically, the authors used an air-temperature index modified by a potential solar radiation index (as affected by slope, aspect, and elevation), canopy cover and ground cover. This modified temperature index was then used to drive a snowpack accounting scheme and a frost depth calculation based on cumulative freezing degree-days.

I found the "frozen ground index" and its relationship to the "modified Berggren Equation" unclear. Is the modified Berggren Equation used only after the frozen ground index exceeds a "threshold" value?

Also, equation (16) on page 6 is puzzling to me. There seems to be an Rnet term (net all-wave radiation) missing. It should read: Rnet = Rsw,net + Rlw_down - Rlw_up. Equation (16) makes sense only if Rnet is assumed to be zero but that is not stated.

It is unclear from the text if sublimation from the snowpack is accounted for when there is no canopy cover. It appears that sublimation is only calculated from intercepted snow.

Page 13, table 3: The term "residual saturation" is ambiguous, particularly at the low value of 0.038. Is this referring to some "degree of saturation", ie, some volumetric soil moisture expressed as a fraction of the saturated volumetric moisture content?

Page 23, line 4. Should read "requires more energy loss to cool and freeze the soil"

Overall, the authors are to be commended for their sound modeling procedures that included several field test sites, distinct calibration versus validation time periods, the sophisticated parameter estimation techniques, and sensitivity analysis. By using a 30-meter grid for capturing spatial variability they were able to generate informative maps of snow depth, frost depth, and snow water equivalent. The use of RMSE and NSE values for each case, as well as plots of absolute simulated and observed values for each time period, were appreciated. The absolute errors were hard to perceive for some data sets simply because of the small size of the graphs. It does appear from the plots that there were cases with high absolute differences between simulated and observed.

I agree with the authors assessment of the model's strengths and weaknesses, and the need for improved representation of the effects of wind and soil moisture in achieving better accuracy.

**3.2 Response to Referee #3**

Main Issue #1, Introduction unnecessarily complicated:

We have revised the introduction to follow a more logical progression. The first paragraph (beginning on Page 1) states that frozen ground is important in estimating stormflow and highlights the heterogeneity of frozen ground and the causes of the spatial variability. The second paragraph describes the current numerical approaches for simulating frozen ground, with energy balance methods often being too complex to use in many watershed modelling scenarios and index-based approaches often not capturing spatial variability outside of elevation. The third and fourth paragraphs focus on the objective of the study, which is to better estimate the spatial pattern of frozen ground in data-sparse watersheds by modifying a commonly-used index-based approach (Lines 30-32 on Page 2).

Main Issue #2, Relationship between frozen ground index and modified Berggren Equation is unclear.:

The modified Berggren Equation (as shown in Equation 19) is used to estimate frost depth from the frozen ground index $F$. The ground freezes when $F$ reaches a threshold $F_{threshold}$, and frost depth increases as $F$ becomes larger than $F_{threshold}$. As $F$ decreases (due to increased temperatures), the frost depth also decreases until no frost is left

when $F$ fall below $F_{threshold}$. We revised Lines 12-16 on Page 8 to better describe the connection between $F$ values calculated by CFGI / modCFGI and the modified Berggren Equation.

Main Issue #3, Net radiation term is missing from Equation 16.:

Equation 11 (formerly Equation 16) represents a step in obtaining the radiation-derived proxy temperature ($T_{rad}$) for simulation of the snowpack. $T_{rad}$ is calculated under the assumption that the outgoing long-wave radiation balances the net incoming short-wave and long-wave radiation (Follum et al., 2015). This balance is clearly an approximation, but the implied proxy temperature better represents the available energy than the air temperature. Follum et al. (2015) showed that the use of $T_{rad}$ can provide a significant improvement in the simulation of snowpack in a temperature index approach. This assumption is now stated directly in Line 30 on Page 5 – Line 6 on Page 6.

Main Issue #4, Unclear on the inclusion of sublimation of the snowpack.:

Like SNOW-17, both the RTI and TI models can crudely account for interception / sublimation /condensation through a precipitation multiplication factor ($S_{cf}$) (Anderson 2006) typically applied uniformly to the watershed (Lines 28-30 on Page 3; and Lines 3-4 on Page 7). The initial manuscript set $S_{cf}$ equal to 1, while calculating canopy sublimation in the RTI model using the common approach described in Liston and Elder (2006). The updated manuscript calibrates $S_{cf}$ in the TI model and in the RTI model uses spatially-varying estimates of sublimation based on solar radiation. The simple method (described in Lines 3-13 on Page 7) estimates sublimation based on the incident solar radiation (and thus land cover and topography). The method neglects wind speed and relative humidity, which affect sublimation rates, but those variables are not available in many cold-region watersheds. Nonetheless, it does vary sublimation rates based on prevalence to solar radiation, which can control sublimation rates and therefore the variability of the snowpack (Gustafson et al., 2010).

Main Issue #5, Ambiguous "residual saturation" term.:

The term "residual saturation" was replaced with "residual water content" in Table 3 on Page 14 and in Lines 9-10 on Page 25.

Minor Comment #1, Change in Line 4 of Page 23:

We appreciate the correction and changed the text from "require more energy to cool and freeze the soil" to "require more energy loss to cool and freeze the soil" in Line 4 on Page 26.

(5)

where $\sigma$ is the Stefan-Boltzmann constant, $f_u$ is the average wind function (mm mb[-1] (6 h)[-1]) (see Anderson (2006) for details), $r_h$ is the relative humidity (assumed to be 0.9 during rain-on-snow events) (Anderson, 1973, 2006), $P_a$ is atmospheric

5  pressure (mb) (either measured or calculated from elevation) (Anderson, 2006), and $e_{sat}$ is the saturation vapor pressure (mb) (calculated based on Smith (1993)). The ripeness of the snowpack affects the amount of melt that is released and is controlled by the liquid holding capacity $L_{hc}$, which is a specified percentage of the ice in the snowpack (Anderson, 2006).

For frozen ground calculations, the snow depth is needed from the snow model. The snow depth $D_s$ (cm) is found from the SWE and the snowpack density. GSSHA uses the single-layer snow density functions from SNOW-17 (Anderson,

10  1976; Anderson, 2006). The density of newly fallen snow $\rho_n$ (gm cm[-3]) varies between 0.05 ($T_a \leq -15°C$) and 0.15 ($T_a = 0°C$) according to:

$$\rho_n = 0.05 + 0.0017\,(T_a + 15)^{1.5}.$$

(6)

Increases in snowpack density $\rho_x$ from compaction, destructive metamorphism, and melt metamorphism due to the presence of liquid water are calculated as (Koren et al., 1999):

15  $$\rho_{x,t} = \rho_{x,t-1}\left(\frac{e^{B_2}}{B_2}\right)e^{B_1},$$

(7)

where:

$$B_1 = c_3\,c_5\,dt\,e^{c_4\,T_s - c_x\,\beta\,(\rho_{x,t-1} - \rho_d)}, \text{ and}$$

(8)

$$B_2 = W_{t-1}\,c_1\,dt\,e^{0.08\,T_s - c_2\rho_{x,t-1}}.$$

(9)

The variable $t$ is an index for time, $W$ is the ice portion of the snow pack (cm, $W = 100\,S_{swe,t-1}$) where $S_{swe}$ is the

20  snow water equivalent on the ground in m, $T_s$ is the average snow pack temperature (°C, calculated based on Anderson (2006)), and $\rho_d$ is the threshold density above which destructive metamorphism decreases ($\rho_d$ is set to 0.15 gm cm[-1] based on Anderson (2006)). Finally, $\beta = 0$ if $\rho_{x,t-1} \leq \rho_d$, and $\beta = 1$ if $\rho_{x,t-1} > \rho_d$, and $c_1$ through $c_5$ are constants (see Anderson (2006) for details).

**2.2 CFGI Frozen Ground Model**

25  The CFGI model was originally developed as a lumped model for flood forecasting in the Pacific Northwest, but it has been used in distributed models as well (De Roo et al., 2001; Van Der Knijff et al., 2010). The rationale of the CFGI method is that air temperature ultimately controls the ground temperature, but its impact is moderated by the insulating effects of any snowpack. The presence of frozen ground is determined by the frozen ground index $F$ (°C-days), which is calculated as:

30  $$F_t = F_{t-1}A - T_{a,d}\,e^{-0.4K_sD_s},$$

(10)

where $T_{a,d}$ is the average daily air temperature (°C), $A$ is a daily decay coefficient, and $K_s$ is the snow reduction coefficient (cm⁻¹). $A$ controls the persistence of the $F$ values, and $K_s$ controls the insulation from the snowpack. Molnau and Bissell (1983) recommended changing $K_s$ depending on whether $T_{a,d}$ is above or below freezing (denoted as $K_{s,T_a>0°C}$ and $K_{s,T_a<0°C}$, respectively).

Higher values of $F$ indicate a higher likelihood that the ground is frozen. Once $F$ exceeds a specified threshold ($F_{threshold}$), the ground is considered frozen and infiltration is restricted. Molnau and Bissell (1983) found the ground to be frozen when $F > 83$ °C-days and thawed when $F < 56$ °C-days. When $F$ is between these values, the ground could be either frozen or thawed. It is worth noting that  $F$ does not depend on soil moisture, which is known to affect the initialization and depth of frozen ground (Kurganova et al., 2007; Willis et al., 1961).

**2.3 RTI Snowpack Model**

The RTI model makes two modifications to the TI model: (1) it  uses a radiation-derived temperature $T_{rad}$ (°C) to better describe the available energy. , and (2) it estimates spatially-varying snowpack sublimation based on solar radiation approximations.

$$P_{eff} = P - I + D, \tag{11}$$

$$I_t = I_{t-1} + 0.0007(I_{max} - I_{t-1})[1 - \exp(-1000\, P/I_{max})], \tag{12}$$

$$D = 1.61 \times 10^{-8}(T_a - 273.16)\Delta t. \tag{13}$$

$$S_{swec,t} = S_{swec,t-1} + I - S_{sub} - D, \tag{14}$$

$$S_{sub} = C_e\, I\, \varphi\, \Delta t, \tag{15}$$

[revised manuscript text omitted]
 more related to the effects of vegetation. In Fig. 3, the two models tend to produce similar results at the pasture sites (see FS24, FS30, and R3), but they tend to differ at the deciduous forest sites (see FS4, FS21, and FS40). At the deciduous sites the RTI model often produces lower peak~~

snowpack due to the interception and sublimation of snowpack by the canopy (which are not included in the TI model). Overall, the RTI model performs better than the TI model (lower RMSE values and higher NSE values in Table 4), suggesting that inclusion of sublimation/interception and use of $T_{raa}$ improve the spatial representation of the snowpack.
[revised manuscript text omitted]

---

## Author Response (AR2)

**A Simple Temperature-Based Method to Estimate Heterogeneous Frozen Ground within a Distributed Watershed Model**

Michael L. Follum[1,2], Jeffrey D. Niemann[2], Julie Parno[3], and Charles W. Downer[1]

[1]Coastal and Hydraulics Laboratory, Vicksburg, MS, 39180, USA.
[2]Department of Civil Engineering, Colorado State University, Fort Collins, CO, 80523, USA.
[3]Cold Regions Research and Engineering Laboratory, Hanover, NH, 03755, USA.

*Correspondence to*: Michael L. Follum (Michael.L.Follum@usace.army.mil)

**Summary of Author Response**

We would like to thank the editor for the suggestions to improve the manuscript. The response from the authors is followed by the "marked-up manuscript". The following response includes the comments from the editor, our response to the comment, the manuscript changes that were made to address the comment, and a reference section. All line and page numbers refer to the "clean" manuscript (with no changes marked).

**1 Comments and Responses**

**1.1 Comments from Editor**

Thank you for submitting the revised version of the manuscript with the detailed response and the changes. In general, I am happy with the revised manuscript and it is close to publication, but I would like to ask the authors to add/correct a couple of minor points before the manuscript can be considered to be published in HESS:

1) Define dt in Eq. 2.

2) why is dt divided by 6 in Eq 2+3 - not clear to me at the moment without coding the equations - if this is an artefact of the time step of thew model, than the equations should we written independent of the time step of the model.

3) as far as I can see, c3, c4, c5 in Eq. 8 are not defined - if they are constant, please provide values.

4) The RTI model does not consider air temperature at all - I know several snow models that combine short wave radiation and air temperature information to improve the model - why is this not done for the RTI model - hence, is

the model than weaker to simulate climate change?

5) I cannot find the Kv values and fu in the equations, but they are important parameters of the model (Tab 1) - please clarify or add.

**1.2 Response to Editor**

1.) $dt$ in Eq. 2, Eq. 3, and Line 6 on Page 4 is now replaced with $\Delta t$ (defined in Line 22 on Page 3).

2.) The SNOW-17 model (Anderson, 1973) was originally based on a 6-hour time step, so many of the model parameters are defined for that time step. Anderson (2006) describes how to adjust model parameters (such as $\Delta D_t$ in Eq.2 and $M_f$ in Eq. 3) for time steps other than 6-hour. Although an artefact of the 6-hour time step originally used in SNOW-17 (and therefore the TI model in this manuscript), the use of "$\Delta t/6$" in Eq. 2 and 3 is how Anderson (2006) developed the equations to be time-step independent while maintaining the original parameter definitions.

3.) The constant parameters related to snow density ($c_1$ through $c_5$) are now defined in Lines 6-8 on Page 5 with reference to Anderson (1976) and Anderson (2006).

4.) Air temperature ($T_a$, °C) is directly included in the RTI model in two ways. First, $T_a$ is used to determine if precipitation events fall as snow or as rain. Second, $T_a$ is used to calculate the downwelling longwave radiation ($R_{LW\downarrow}$, W m$^{-2}$), which is used to calculate $T_{rad}$ (°C) in Eq. 12. In regards to climate change applications (e.g. shifts in average air temperature), the RTI model would capture the difference between rainfall and snowfall events as well as increased/decreased available energy due to increased/decreased values of $T_a$.

5.) The vegetation transmission coefficient $K_v$ is substituted for $\varphi_v$ in Eq. 14, which is now better defined in Lines 17-19 on Page 6. The average wind function $f_u$ is used in Eq. 5 for rain-on-snow melt events.

[revised manuscript text omitted]